# Activation of the Keap1/Nrf2 pathway suppresses mitochondrial dysfunction, oxidative stress, and motor phenotypes in *C9orf72* ALS/FTD models

Wing Hei Au[1,2] ⓘ, Leonor Miller-Fleming[1,]* ⓘ, Alvaro Sanchez-Martinez[1,]* ⓘ, James AK Lee[3,]* ⓘ, Madeleine J Twyning[1] ⓘ, Hiran A Prag[1,4] ⓘ, Laura Raik[1], Scott P Allen[3] ⓘ, Pamela J Shaw[3,5], Laura Ferraiuolo[3] ⓘ, Heather Mortiboys[3] ⓘ, Alexander J Whitworth[1] ⓘ

**Mitochondrial dysfunction is a common feature of *C9orf72* amyotrophic lateral sclerosis/frontotemporal dementia (ALS/FTD); however, it remains unclear whether this is a cause or consequence of the pathogenic process. Analysing multiple aspects of mitochondrial biology across several *Drosophila* models of *C9orf72*-ALS/FTD, we found morphology, oxidative stress, and mitophagy are commonly affected, which correlated with progressive loss of locomotor performance. Notably, only genetic manipulations that reversed the oxidative stress levels were also able to rescue *C9orf72* locomotor deficits, supporting a causative link between mitochondrial dysfunction, oxidative stress, and behavioural phenotypes. Targeting the key antioxidant Keap1/Nrf2 pathway, we found that genetic reduction of *Keap1* or pharmacological inhibition by dimethyl fumarate significantly rescued the *C9orf72*-related oxidative stress and motor deficits. Finally, mitochondrial ROS levels were also elevated in *C9orf72* patient-derived iNeurons and were effectively suppressed by dimethyl fumarate treatment. These results indicate that mitochondrial oxidative stress is an important mechanistic contributor to *C9orf72* pathogenesis, affecting multiple aspects of mitochondrial function and turnover. Targeting the Keap1/Nrf2 signalling pathway to combat oxidative stress represents a therapeutic strategy for *C9orf72*-related ALS/FTD.**

## Introduction

Amyotrophic lateral sclerosis (ALS) is characterised by the loss of upper and lower motor neurons leading to symptoms such as muscle weakness and paralysis. A plethora of evidence supports a clinical, pathologic, and genetic overlap between ALS and frontotemporal dementia (FTD) [1], which is characterised by the degeneration of frontal and temporal lobes leading to clinical symptoms such as cognitive impairment and changes in behaviour and personality. Up to 50% of ALS patients show cognitive and behavioural changes; similarly, motor neuron dysfunction is a common feature in ~15% of FTD cases [2]. A hexanucleotide repeat expansion consisting of GGGGCC (G4C2) in the first intron of *C9orf72* is the most common pathogenic mutation in ALS/FTD [3, 4]. Several pathogenic mechanisms have been proposed including haploinsufficiency of the gene product and the sequestration of RNA-binding proteins at accumulations (foci) of the transcribed RNA [5]. Although intronic, the expanded RNA can also be translated through a mechanism known as repeat-associated non-AUG translation, which produces five different dipeptide repeat proteins (DPRs), with arginine DPRs exhibiting the most severe toxicity [6, 7, 8, 9, 10, 11, 12, 13, 14].

Abundant evidence supports that mitochondrial dysfunction is an early alteration in ALS [15, 16], and mitochondrial bioenergetics and morphological changes have been observed in *C9orf72* patient fibroblasts and iPSC-derived motor neurons [17, 18, 19]. Cellular and mouse models expressing poly-GR have consistently shown mitochondrial perturbations such as redox imbalance and DNA damage [20]. Evidence indicates that poly-GR binds to ATP5A1, thereby compromising mitochondrial function [21], and that poly-GR toxicity and aggregation can also occur due to frequent stalling of poly-GR translation on the mitochondrial surface, triggering ribosome-associated quality control and C-terminal extension [22].

Various cellular mechanisms exist to counteract disruptions in mitochondrial function and redox imbalance, from antioxidant defence mechanisms to the wholesale degradation of

---

[1]MRC Mitochondrial Biology Unit, University of Cambridge, Cambridge, UK   [2]John van Geest Centre for Brain Repair, Department of Clinical Neurosciences, University of Cambridge, Cambridge, UK   [3]Sheffield Institute for Translational Neuroscience (SITraN), School of Medicine and Population Health, University of Sheffield, Sheffield, UK   [4]Department of Medicine, University of Cambridge, Cambridge, UK   [5]NIHR Sheffield Biomedical Research Centre, Sheffield Teaching Hospitals NHS Foundation Trust, Sheffield, UK

Correspondence: a.whitworth@mrc-mbu.cam.ac.uk
*Leonor Miller-Fleming, Alvaro Sanchez-Martinez, and James AK Lee contributed equally to this work

---

mitochondria via macroautophagy (mitophagy), which has been relatively understudied in the *C9orf72* context. One of the most important upstream mechanisms that counteracts redox imbalance is the Keap1/nuclear factor erythroid 2–related factor 2 (Nrf2) pathway. Under basal conditions, Nrf2, a master regulatory transcription factor for antioxidant and cell-protective factors, is negatively regulated via targeted proteasomal degradation by an E3 ubiquitin ligase adaptor, Keap1. This interaction is alleviated upon oxidative and electrophilic stresses and can also be provoked by Keap1 small-molecule inhibitors, causing Nrf2 to accumulate in the nucleus. This induces the expression of a repertoire of protective factors such as proteins with detoxification, antioxidant, and anti-inflammatory properties, with the purpose of maintaining mitochondrial function and redox balance (23, 24).

In this study, we have conducted an extensive in vivo characterisation of mitochondrial dysfunction in multiple *Drosophila* models of *C9orf72* ALS/FTD by studying mitochondrial dynamics, respiration, mitophagy, and redox homeostasis. We found that only the reversal of oxidative stress by the overexpression of antioxidant genes was able to rescue the progressive loss of motor function. Focussing on a role of the key antioxidant Keap1/Nrf2 signalling pathway, we investigated genetic and pharmacological inactivation of the negative regulator Keap1 in the *C9orf72 Drosophila* models and *C9orf72* patient-derived iNeurons. Our results suggest that mitochondrial oxidative stress is an upstream pathogenic mechanism and activation of the Keap1/Nrf2 pathway could be a viable therapeutic strategy for ALS/FTD.

## Results

### *C9orf72 Drosophila* models present a wide range of mitochondrial defects

Mitochondrial dysfunction has been suggested to be directly involved in disease pathogenesis in ALS/FTD, but evidence from animal models is limited (25). We sought to investigate this in vivo using *Drosophila* as a model system. *Drosophila* do not encode an orthologue of *C9orf72*; nevertheless, exogenous genes can be readily expressed in *Drosophila* based on the inducible GAL4-UAS expression system (26). Several groups have previously established different *Drosophila* models of *C9orf72*-related pathology, which express pure hexanucleotide repeats or alternative codon DPRs of various lengths, sometimes with additional genomic elements of *C9orf72* (27). We focussed our initial efforts on readily available lines that express pure G4C2 repeats or alternative codon repeats that encode the DPRs. The expression of 36x G4C2 repeats (G4C2x36) and 36-repeat glycine–arginine DPR (GR36) is substantially neurotoxic compared with the 3x G4C2 repeat (G4C2x3) control (6). Indeed, we found that the strong pan-neuronal expression of G4C2x36 (via *nSyb*-GAL4) perturbed development, leading to larvae morphologically thinner compared with controls (Fig 1A), and with significantly reduced motor ability assessed by measuring larval

crawling behaviour (Fig 1B). Consistent with previous reports, the pan-neuronal expression of GR36 caused even stronger phenotypes (Fig 1A and B), and was developmentally lethal at the third-instar (L3) larval stage.

To analyse *C9orf72* pathology in the adult stage, where the impact of ageing can also be investigated, we found that the expression of the *C9orf72* transgenes via a predominantly motor neuron driver, *DIPγ*-GAL4, was adult-viable and caused an age-related decline in motor ability assessed using the negative geotaxis "climbing" assay (Fig 1C). While the motor neuron expression of G4C2x36 did not impact motor behaviour at 2–d-old, by 10 d, G4C2x36 flies exhibited significantly impaired climbing ability (Fig 1C). As before, the motor neuron expression of GR36 was more toxic, causing a significant motor deficit at 2-d-old, which worsened by 10 d (Fig 1C). A similar pattern was observed in the lifespan of *DIPγ*-GAL4–driven G4C2x36 or GR36 flies with a mild phenotype from G4C2x36 but a markedly shortened lifespan with GR36 (Fig S1A).

Previous studies analysing mitochondrial defects in *Drosophila C9orf72* models have mostly focussed on the muscle-directed expression of poly-GR (22, 28). Here, we aimed to explore the potential involvement of mitochondrial dysfunction in a neuronal context. Initially, we observed that the pan-neuronal expression of G4C2x36 caused a significant reduction in complex I– and complex II–linked respiration in lysates from young (5-d-old) fly heads (Fig 1D), indicative of a generalised mitochondrial disruption. To probe this in more detail, we next analysed mitochondrial morphology in larval neurons upon the pan-neuronal expression of the *C9orf72* transgenes. As mitochondria are dynamic organelles, microscopy analysis of mito.GFP-labelled mitochondria typically reveals a mix of short round (fragmented) and long tubular (fused) morphologies (Fig 1E), which can be quantified using a scoring system that characterises the overall mitochondrial morphology on a cell-by-cell basis (Fig 1E and F). Using this approach, we found that mitochondria were more elongated and hyperfused in G4C2x36- and GR36-expressing neurons (Fig 1E and F).

Mitochondrial morphology is known to respond to changes in reactive oxygen species (ROS) levels, as well as other physiological stimuli. First, we tested the susceptibility of G4C2x36 flies to oxidative stress induced by paraquat (which generates superoxide anions) and hydrogen peroxide ($H_2O_2$, which generates hydroxyl radicals). We observed that G4C2x36 flies were hypersensitive to both types of oxidative stressors when compared to control animals (Fig S1B and C).

ROS can occur in different forms (e.g., superoxide anions, $H_2O_2$), which may vary in their prevalence and functional significance across cellular compartments (mitochondrial versus cytosolic). Therefore, it is important to study the effects of ROS species in each compartment to understand the tight regulation required to achieve redox homeostasis. We used genetically encoded redox-sensitive fluorescent protein (roGFP2) probes fused to glutaredoxin-1 (Grx1) or oxidant receptor peroxidase 1 (Orp1). Oxidation by oxidised glutathione (GSSG) or $H_2O_2$, respectively, leads to a shift in the excitation maxima of the roGFP2 fusion constructs from 488 to 405 (29). Importantly, these

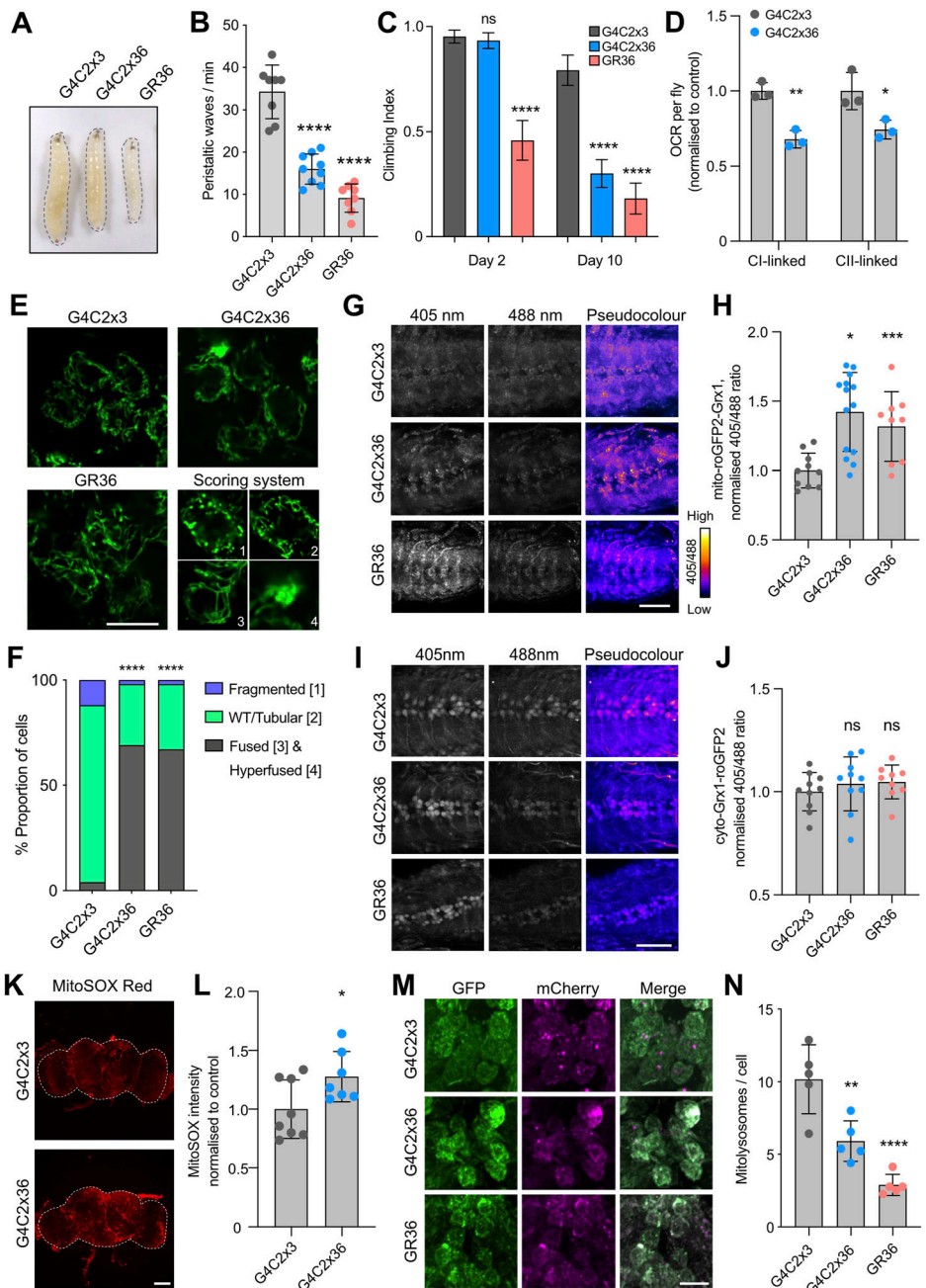

**Figure 1. Multiple aspects of mitochondrial function are disrupted in *Drosophila C9orf72* models.**

**(A, B)** Morphology and (B) locomotor ability (crawling) of larvae expressing G4C2x3, G4C2x36, or GR36 via a pan-neuronal driver *nSyb*-GAL4 (mean ± SD; one-way ANOVA with Bonferroni's multiple comparison test, ****$P < 0.0001$; n = 8–10 larvae). **(C)** Climbing analysis of 2- and 10-d-old adults expressing G4C2x3, G4C2x36, or GR36 flies via a predominantly motor neuron driver *DIPγ*-GAL4 (mean ± 95% CI; Kruskal–Wallis non-parametric test with Dunn's correction, ****$P < 0.0001$; n = 55–100 flies). **(D)** Oxygen consumption rate of 5-d-old heads expressing G4C2x3 or G4C2x36 using *nSyb*-GAL4 (mean ± SD; unpaired *t* test with Welch's corrections, *$P < 0.05$, **$P < 0.01$; n = 3 biological replicates). **(E)** Confocal microscopy of larval neurons where mitochondria are labelled with the pan-neuronal expression of mito.GFP, co-expressing G4C2x3, G4C2x36, or GR36 with *nSyb*-GAL4. Scale bar = 10 $\mu$m. **(E, F)** Quantification of (E) on a cell-by-cell basis as (1) fragmented, (2) tubular (WT) appearance, (3) fused, or (4) hyperfused (chi-squared test, ****$P < 0.0001$; n = 8–10 larvae). **(G, H, I, J)** Confocal microscopy of the (G) mito-roGFP2-Grx1 mitochondrial glutathione redox potential reporter and (I) cyto-Grx1-roGFP2 cytosolic glutathione redox potential reporter co-expressed with G4C2x3, G4C2x36, or GR36 in the larval ventral ganglion with *nSyb*-GAL4. Representative pseudocolour images of the 405/488 ratio are shown; scale bar = 50 $\mu$m. **(H, J)** Quantification of the 405/488 ratio analysed in G and I, respectively (mean ± SD; one-way ANOVA with Bonferroni's multiple comparison test, *$P < 0.05$, ***$P < 0.001$, ns = non-significant; n = 9–14 larvae). **(K, L)** MitoSOX staining in 5-d-old G4C2x3 or G4C2x36 brains (pan-neuronal expression with *nSyb*-GAL4). The white dotted line indicates brain boundaries. Scale bar = 100 $\mu$m (mean ± SD; unpaired *t* test with Welch's corrections, *$P < 0.05$; n = 7–8 flies). **(M, N)** Confocal analysis of the mito-QC mitophagy reporter co-expressed with G4C2x3, G4C2x36, or GR36 in the larval ventral ganglion with *nSyb*-GAL4. Scale bar = 10 $\mu$m (mean ± SD; one-way ANOVA with Bonferroni's multiple comparison test, **$P < 0.01$, ****$P < 0.0001$; n = 5 larvae).

reporters can be targeted to specific subcellular compartments, that is, cytosol or mitochondria, giving insights into the compartment-specific redox status. We observed a significant increase in the oxidised status of mitochondrial roGFP2-Grx1 (Fig 1G and H) and mitochondrial roGFP2-Orp1 (Fig S1D and E) in larval neurons when combined with G4C2x36 or GR36, indicating an increase in mitochondrial $H_2O_2$ and GSSG in these animals. In contrast, no significant differences were observed in cytosolic glutathione redox potential ($E_{GSH}$) via cyto-Grx1-roGFP2 (Fig 1I and J), and little or no changes were detected with dihydroethidium (DHE) intensity, an indicator of cytosolic

superoxide, in G4C2x36 and GR36 larvae (Fig S1F and G). To extend our analysis of the larval model to adult flies, and to investigate an independent measure of ROS, we used MitoSOX to investigate levels of mitochondrial superoxide in the adult brain. In agreement with the fluorescent reporter results, we observed an increase in MitoSOX intensity in 5-d-old G4C2x36 flies compared with the G4C2x3 control (Fig 1K and L). Taken together, these data suggest that the mitochondrial but not cytosolic redox state is disrupted in these *Drosophila* models of *C9orf72*-related pathology. These results are summarised in Table 1.

**Table 1. Summary of ROS outcomes in G4C2x36 flies.**

| | | ROS species | | |
| --- | --- | --- | --- | --- |
| | | Superoxide | Hydrogen peroxide | $E_{GSH}$ |
| Compartment | Mitochondrial | MitoSOX[a] (↑) | mito-roGFP2-Orp1 reporter (↑) | mito-roGFP2-Grx1 reporter (↑) |
| | Cytosolic | DHE (−) | n/a[b] | cyto-Grx1-roGFP2 reporter (−) |

KEY: (↑), increased ROS species; (−), no change
[a]Tested in both G4C2x36 and GR1000 flies.
[b]Tried cyto-roGFP2-Orp1 reporter but was too weak in our hands.

Changes in mitochondrial morphology and increases in ROS can be indicative of ongoing mitochondrial damage, which can be alleviated via mitophagy. Thus, we next used the mito-QC mitophagy reporter (30) to investigate the mitophagy status in the *C9orf72* models. Briefly, the mito-QC reporter uses a tandem GFP-mCherry fusion protein targeted to the outer mitochondrial membrane. Mitolysosomes are marked when GFP is quenched by the acidic environment but mCherry fluorescence is retained, resulting in "red-only" puncta (30). Analysis showed that there were significantly fewer mitolysosomes in G4C2x36 and GR36 larval neurons compared with the G4C2x3 control (Fig 1M and N), suggesting that mitophagy is perturbed. However, this was not a mitophagy-specific defect as the analogous reporter for general autophagy, GFP-mCherry-Atg8a (fly Atg8a is homologous to LC3), also showed a reduction in the number of mCherry-positive autolysosomes in G4C2x36 and GR36 larval neurons (Fig S1H and I). Consistent with this, immunoblot analysis of protein lysates from 5-d-old G4C2x36 fly heads revealed an increase in ref(2)P levels (fly homologue of p62), as well as a reduction in lipidated Atg8a-II levels (Fig S1J and K). These results are consistent with a reduction in general autophagic flux, which could impact mitophagic flux, in agreement with previous literature using different *C9orf72* models (31).

Recently, West et al (32) developed additional *C9orf72 Drosophila* models, which express more physiologically relevant repeat length (27), that is, ~1,000 repeats. For comparison with the Mizielinska et al lines, we also analysed the GR1000-eGFP line (for simplicity, hereafter called GR1000). We observed an age-related decline in motor performance compared to control with the pan-neuronal

expression of GR1000, where GR1000 flies were not able to climb at all by 20 d (Fig 2A), similar to that previously reported (32). To complement our results shown in the G4C2x36 and GR36, we measured mitochondrial superoxide with MitoSOX staining in GR1000 adult brains as it was not possible to use the roGFP2 reporters in conjunction with eGFP-tagged GR1000. Here, we also observed an increase in MitoSOX staining in 10-d-old GR1000 brains compared with the control (Fig 2B and C).

Taken together, the preceding data show that various aspects of mitochondrial form and function are disturbed in multiple *Drosophila* models of *C9orf72* pathology, which correlate with the organismal decline and loss of motor function.

### Genetic manipulation of mitochondrial dynamics or mitophagy does not rescue *C9orf72* phenotypes

We next sought to determine whether disruptions to these observed changes in mitochondrial morphology, defective mitophagy, and oxidative stress are contributing factors to the neurodegenerative process. First, we evaluated whether genetic manipulations to counteract the elongated, hyperfused mitochondrial morphology caused by G4C2x36 or GR36 expression may rescue the larval locomotor defect. Thus, we combined the pan-neuronal expression of G4C2x36 or GR36 with genetic manipulations to promote fission (overexpression of pro-fission factors *Drp1* or *Mff* [*Tango11* in *Drosophila*]) or reduce fusion (loss of pro-fusion factors *Opa1* and *Mitofusin* [*Marf* in *Drosophila*]). Although the elongated mitochondria observed in the G4C2x36 and GR36 neurons were partially

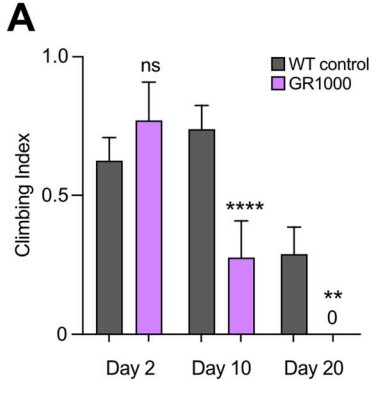
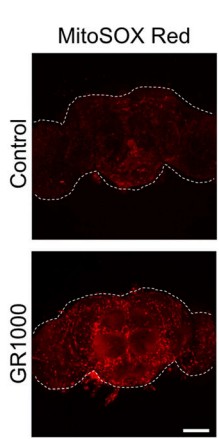
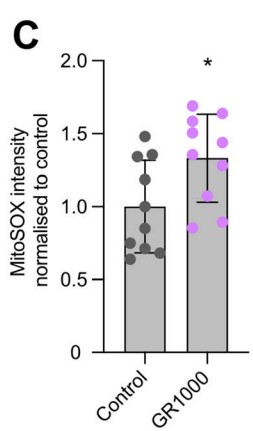

**Figure 2. GR1000 flies show increased mitochondrial ROS.**
**(A)** Climbing analysis of 2-, 10-, and 20-d-old adults of the pan-neuronal expression, via *nSyb*-GAL4, of mito.GFP (WT control) or GR1000 (mean ± 95% CI; Kruskal–Wallis non-parametric test with Dunn's correction, ns = non-significant, **P < 0.01, ****P < 0.0001; n = 60–100 flies). **(B, C)** Confocal microscopy and quantification of MitoSOX staining in 10-d-old adult brains in WT control (*nSyb>mito.GFP*) and GR1000 flies (pan-neuronal expression with *nSyb*-GAL4). White dotted line indicates adult brain boundaries. Scale bar = 100 μm (mean ± SD; unpaired t test with Welch's corrections, *P < 0.05; n = 10 brains).

reversed by these manipulations as expected (Fig S2A and B), this did not result in any improvement in larval locomotion (Fig S2C). These data suggest that excess mitochondrial fusion observed in the *C9orf72* models does not play a key role in *C9orf72* pathogenesis and may be a downstream consequence.

Similarly, since we observed reduced mitophagy in the *C9orf72* models, we aimed to reverse this by boosting mitophagy by targeting the mitophagy inhibitor USP30. Knockdown of *USP30* has been shown to rescue defective mitophagy caused by pathogenic mutations in *PRKN* and improve mitochondrial integrity in *parkin-* or *Pink1*-deficient flies (33). We have also seen that *USP30* knockdown increases mitolysosome number in *Drosophila* neurons and muscle (34), indicative of increased mitophagy. As expected, we observed that *USP30* knockdown significantly increased the number of mitolysosomes in control larval neurons (Fig S2D and E). However, when co-expressed with G4C2x36 or GR36, *USP30* knockdown was not sufficient to rescue the reduced mitophagy (Fig S2D and E). Moreover, the pan-neuronal co-expression of G4C2x36 or GR36 with *USP30* RNAi did not improve the larval locomotor deficit either (Fig S2F). These data suggest that mitophagy is also not a primary contributor to *C9orf72* pathogenesis.

### Overexpression of mitochondrial *Sod2* and *catalase* ameliorates *C9orf72* motor phenotypes

Previous studies have shown that targeting oxidative stress may be beneficial in *C9orf72*-related pathology (20). Cellular defence mechanisms such as antioxidants are targeted to different cellular and subcellular locations, due to many sources of ROS. This compartmentalisation also highlights the need for fine-tuning of ROS signalling for redox homeostasis, as well as the possibility for ROS to signal between compartments (35). Since an increase in mitochondrial ROS was observed in G4C2x36, GR36, and GR1000 flies, we hypothesised that the overexpression of antioxidants would suppress behavioural locomotor phenotypes. First, we used a pan-neuronal driver to co-express the major antioxidant genes—cytosolic *Sod1*, mitochondrial *Sod2*, *catalase* (*Cat*), and a mitochondrially targeted catalase (*mitoCat*)—with G4C2x36 and GR36. Contrary to our prediction, the overexpression of cytosolic *Sod1* significantly worsened the morphology of G4C2x36 and GR36 larvae, becoming even thinner and more developmentally delayed (Fig 3A). This was reflected by a worsening of the locomotor deficit for G4C2x36 and GR36 larvae, where GR36 larvae co-expressing *Sod1* did not crawl during the assay conditions (Fig 3B). In contrast, the overexpression of mitochondrial *Sod2*, *Cat*, or *mitoCat* significantly rescued G4C2x36 and GR36 larval crawling (Fig 3B). These manipulations also improved larval morphology, particularly for GR36 (Fig 3A). Interestingly, we found that the expression of *Sod1* was reduced at both mRNA (Fig S2G) and protein levels (Fig S2H and I) in G4C2x36 flies, whereas *Cat* mRNA levels were increased and *Sod2* mRNA levels were unaltered (Fig S2G).

We next asked whether these manipulations similarly affected the other mitochondrial phenotypes. We found that overexpressing *Sod2*, *Cat*, or *mitoCat* significantly reduced the elongated mitochondrial phenotype in G4C2x36 and GR36 larval neurons (Fig 3C and D). Interestingly, *Sod1* overexpression also partially rescued

this phenotype. Notably, *Sod2* overexpression was also able to partially rescue the defective mitophagy in G4C2x36 larvae, though this did not reach significance for GR36 (Fig 3E and F). To validate these genetic interactions, we used the GR1000 lines where we also found that *Sod2* or *Cat* overexpression significantly rescued the age-related locomotion defect (Fig 3G and H).

Thus, among all the different genetic manipulations used to reverse mitochondrial phenotypes observed, only the overexpression of certain antioxidant enzymes such as mitochondrial *Sod2* and *catalase* was consistently beneficial in rescuing the locomotor deficits. This suggests that oxidative stress is an important mediator of pathogenesis.

### The Keap1/Nrf2 pathway is partially activated in *C9orf72* flies

Keap1 and Nrf2 define an important redox-sensing pathway which activates antioxidant gene expression upon oxidative stress. *Keap1* and *Nrf2* are conserved in *Drosophila* (*Nrf2* is called *cap-n-collar* C isoform, CncC) (36), so we sought to explore whether this pathway is involved in *C9orf72* pathology. Assessing the relative nuclear abundance of CncC in neurons of the larval ventral ganglion, we observed an increase in nuclear CncC in both G4C2x36 and GR36 compared with the G4C2x3 control (Fig 4A and C). We also observed a similar increase in CncC nuclear staining in 5-d-old adult brains of G4C2x36 animals (Fig 4B and C). This indicates that there is activation of the Keap1/CncC signalling pathway at early stages, which remains activated during disease progression.

To further analyse the activity of this signalling pathway, we used an antioxidant reporter transgene, which expresses GFP via the promoter of *GstD1*, an Nrf2 target and prototypical oxidative stress response gene (36), to monitor antioxidant responses. The pan-neuronal expression of both G4C2x36 (Fig 4D) and GR1000 (Fig 4E) caused an increase in GstD1-GFP levels compared with their respective controls, consistent with an increase in CncC activity. At the same time, we assessed whether G4C2x36 flies could also respond appropriately to ROS-inducing paraquat (PQ) treatment. We again saw that the GstD1-GFP reporter expression was significantly increased in G4C2x36 flies under basal conditions (−PQ) compared with controls (Fig 4F). However, although GstD1-GFP levels substantially increased upon PQ treatment in both control and G4C2x36 flies (Fig 4F), the response in G4C2x36 flies was similar to the control. These observations, together with the preceding genetic studies, suggest that although the *C9orf72* flies can detect the elevated oxidative stress conditions and mount a response, it is not maximally induced and likely insufficient to confer full protection. The increased response of the GstD1-GFP reporter to PQ beyond the level seen in G4C2x36 alone indicates that an increased antioxidant response can be induced in these animals.

### Genetic and pharmacological inhibition of Keap1 partially rescues *C9orf72* phenotypes

Although the Keap1/Nrf2 pathway was activated in *C9orf72* models, the response seemed to be insufficient. Therefore, we hypothesised that genetically reducing Keap1 levels to constitutively boost CncC activity could benefit *C9orf72* phenotypes as has been seen in a

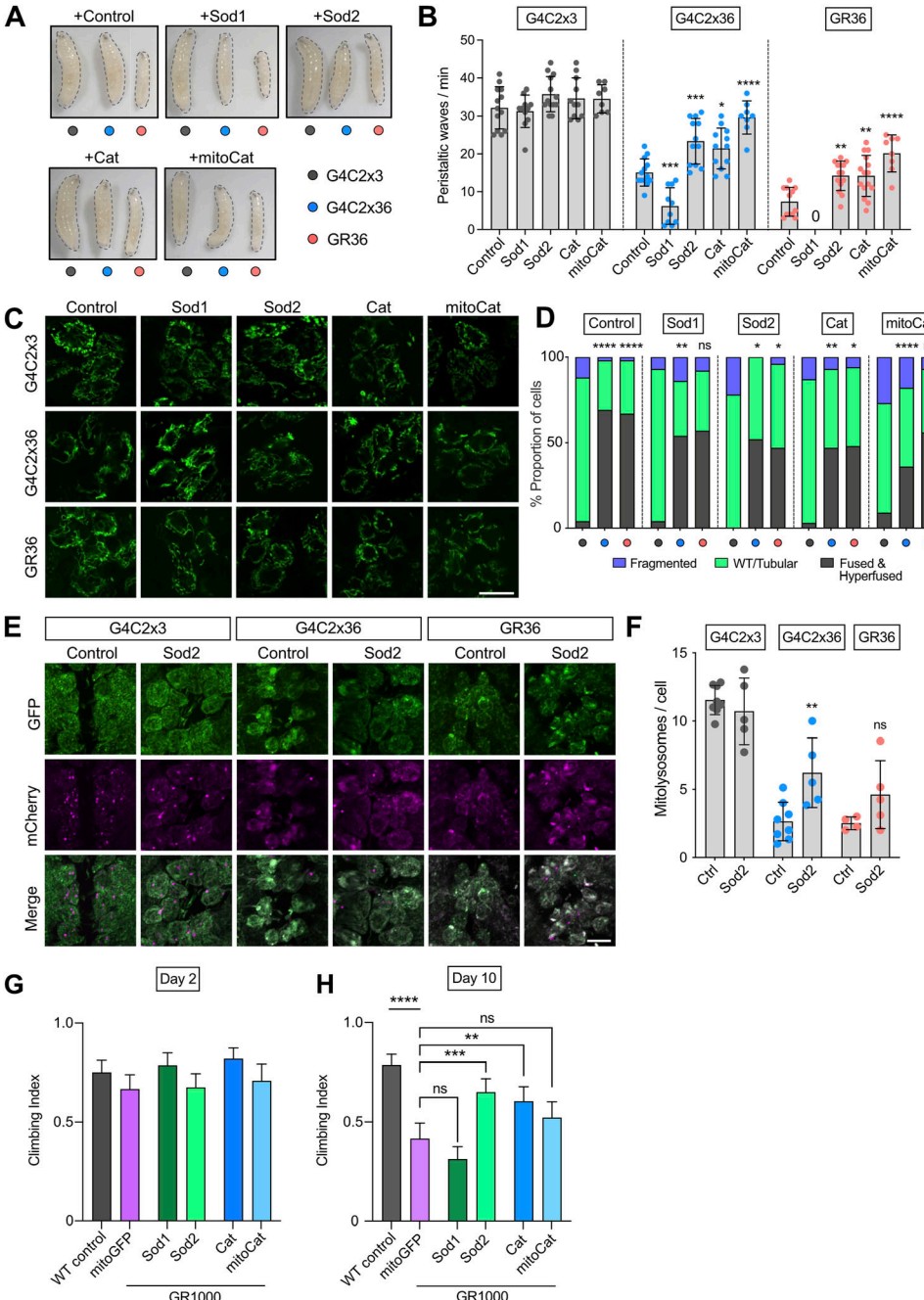

Figure 3. Overexpression of mitochondrial Sod2 and catalase partially rescues *C9orf72* phenotypes.
**(A, B)** Morphology and (B) locomotor ability (crawling) of larvae co-expressing G4C2x3, G4C2x36, or GR36 with *LacZ* (control), *Sod1*, *Sod2*, *catalase* (*Cat*), or mitochondrially targeted *catalase* (*mitoCat*) with a pan-neuronal driver *nSyb*-GAL4 (mean ± SD; one-way ANOVA with Bonferroni's multiple comparison test, *P < 0.05, **P < 0.01, ***P < 0.001, ****P < 0.0001; 0 denotes larvae that did not crawl; n = 10–15 larvae); scale bar = 10 μm. **(C)** Confocal microscopy of larval neurons. Mitochondria are labelled with the pan-neuronal expression of mito.GFP, co-expressing G4C2x3, G4C2x36, or GR36 with *nSyb*-GAL4. Genetic manipulations boosting antioxidant capacities by overexpressing *Sod1*, *Sod2*, *Cat*, or *mitoCat*. Control is overexpressing *LacZ*. Scale bar = 10 μm. **(C, D)** Quantification of (C) based on the established scoring system (chi-squared test, ns = non-significant, *P < 0.05, **P < 0.01, ****P < 0.0001; n = 8–10 larvae). Comparisons for the first *LacZ* group are against G4C2x3, *LacZ*. Otherwise, all the comparisons are against their respective control (*LacZ*) conditions. **(E, F)** Confocal microscopy and (F) quantification of the mito-QC mitophagy reporter of larvae co-expressing G4C2x3, G4C2x36, or GR36 with luciferase RNAi (control) and *Sod2* in the larval ventral ganglion with *nSyb*-GAL4. Scale bar = 10 μm. Mitolysosomes are evident as GFP-negative, mCherry-positive puncta (mean ± SD; one-way ANOVA with Bonferroni's multiple comparison test, ns = non-significant, **P < 0.01; n = 4–8 larvae). **(G, H)** Climbing analysis of 2- and 10-d-old adults expressing GR1000 co-expressing *mito.GFP*, *Sod1*, *Sod2*, *Cat*, and *mitoCat* with *nSyb*-GAL4. WT control is *nSyb/+* (mean ± 95% CI; Kruskal–Wallis non-parametric test with Dunn's correction, ns = non-significant, **P < 0.01, ***P < 0.001, ****P < 0.0001; n = 60–100 flies).

*Drosophila* model of Alzheimer's disease (37). Indeed, combining a heterozygous mutant of *Keap1* with pan-neuronally driven G4C2x36 or GR36 significantly improved the larval motor ability of both G4C2x36 and GR36 compared with the control (Fig 5A), and visibly improved the larval morphology (Fig 5B). Furthermore, *Keap1* heterozygosity was also able to fully rescue the GR1000 adult climbing deficit (Fig 5C).

Analysing the distribution of CncC, we observed again that G4C2x36 provoked an increase in CncC nuclear localisation (Fig 5D and E) with the trend towards a further increase with *Keap1*

heterozygosity (Fig 5D and E). Similarly, when performing high-resolution respirometry, there was a trend for improving both complex I– and complex II–linked respiration deficit in G4C2x36 with *Keap1* heterozygosity (Fig 5F).

The positive effects of partial genetic loss of *Keap1* on the behaviour of *C9orf72* models support the potential for pharmaceutical agents that modulate Keap1 to also be beneficial in this context. To test this hypothesis, we used dimethyl fumarate (DMF), a Keap1-modifying Nrf2-activating drug with antioxidative and anti-inflammatory properties. DMF acts by inactivating Keap1 via

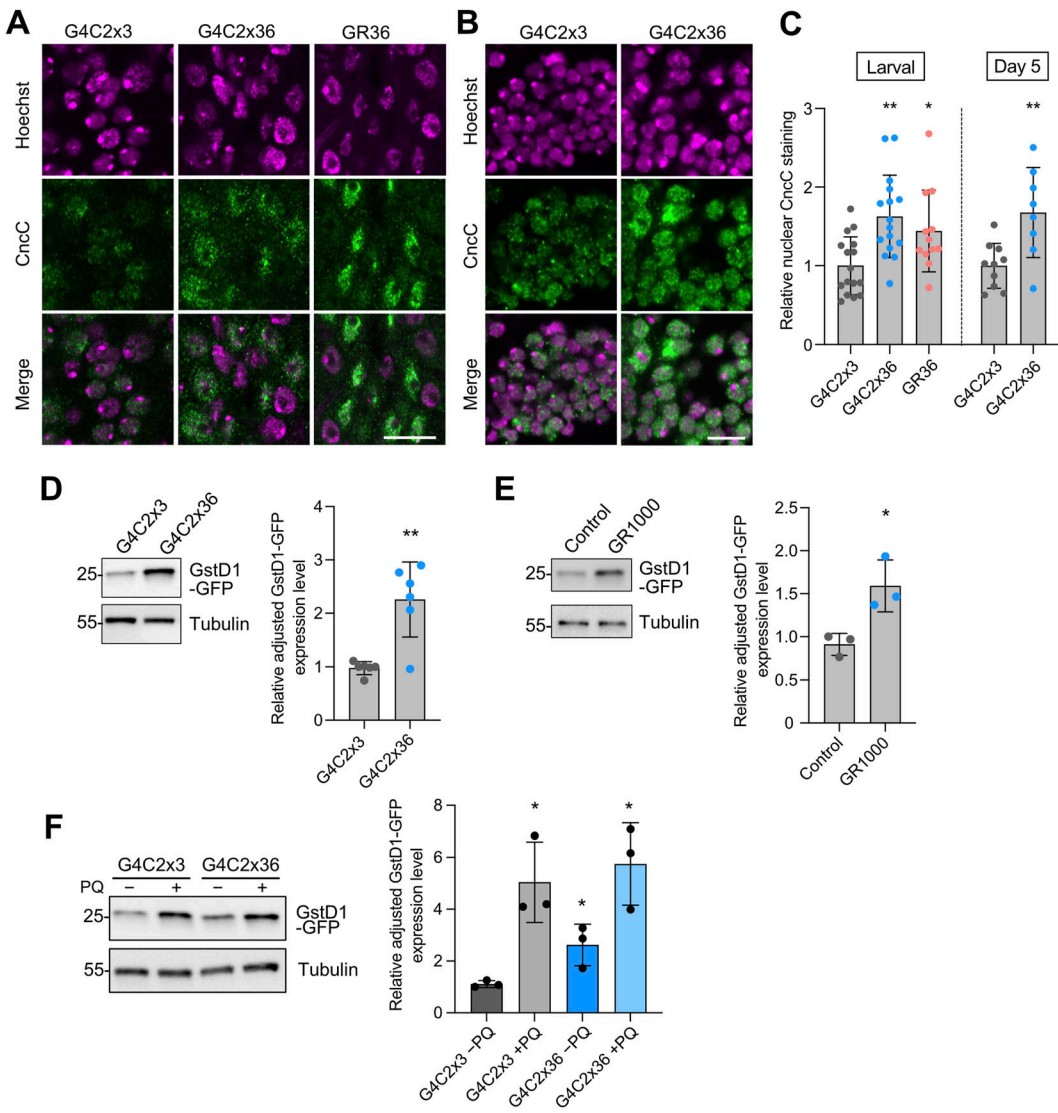

**Figure 4. Keap1/CncC pathway is partially activated in *C9orf72* pathogenesis.**
**(A, B)** Confocal microscopy images of larval ventral ganglion and (B) 5-d-old adult brains immunostained for CncC expression, with Hoechst used to delineate the nuclei. The pan-neuronal expression of G4C2x36 or GR36 was via *nSyb*-GAL4. **(C)** Quantification of relative nuclear CncC staining normalised to the control, G4C2x3 (mean ± SD; one-way ANOVA with Bonferroni's multiple comparison test for larval brain analysis, *$P < 0.05$, **$P < 0.01$; n = 12–18 larval brains; unpaired $t$ test with Welch's correction for adult brain analysis, **$P < 0.01$; n = 8-10 adult brains). **(D, E)** Immunoblot and quantification of *GstD1-GFP* reporter transgene levels (via anti-GFP), in adult brain samples of the pan-neuronal expression of (D) G4C2x3 or G4C2x36 at day 5, and (E) LacZ (control) and GR1000 at day 10. Tubulin was used as the loading control (mean ± SD; unpaired $t$ test with Welch's correction, *$P < 0.05$, **$P < 0.01$; n = 3–6 biological replicates). **(F)** Immunoblot and quantification of *GstD1-GFP* reporter transgene levels in 5-d-old adult brain samples of the pan-neuronal expression of G4C2x3 or G4C2x36, with and without 5 mM PQ treatment (mean ± SD; one-way ANOVA with Bonferroni's correction, *$P < 0.05$; n = 3 biological replicates).

succination of its cysteine residues (38), which has previously been shown to exhibit neuroprotective effects in animal models of neurodegeneration (39). Analysing the impact of DMF on the Keap1/Nrf2 pathway, we performed quantitative RT–PCR on transcriptional targets of CncC, including *GstD1*, *GstD2*, *Gclc*, and *Cyp6a2* (40, 41), in the *C9orf72* models with and without a modest dose of DMF treatment. The degree of transcriptional activation was variable between different targets, but DMF treatment caused a general up-regulation of CncC targets in G4C2x36 flies (Fig 6A).

To test whether DMF treatment improves mitochondrial function in the G4C2x36 model, we again used the mito-roGFP2-Orp1 reporter

to assess mitochondrial $H_2O_2$ levels. Here, animals were raised on food containing DMF or vehicle (ethanol). We again observed an increase in reported $H_2O_2$ levels in G4C2x36 animals, which was partially rescued by DMF treatment (Fig 6B).

Finally, to extend these observations and assess the effects of DMF treatments on motor phenotypes, we expressed G4C2x3, G4C2x36, or GR36 via *DIPγ-GAL4* and treated the adult flies with food supplemented with DMF or vehicle (ethanol) for 10 d. For both G4C2x36 and GR36, the progressive decline in locomotor performance in vehicle-treated flies was greatly alleviated by 10 d of DMF treatment (Fig 6C). Similarly, the progressive loss of motor function in pan-neuronal GR1000 flies was

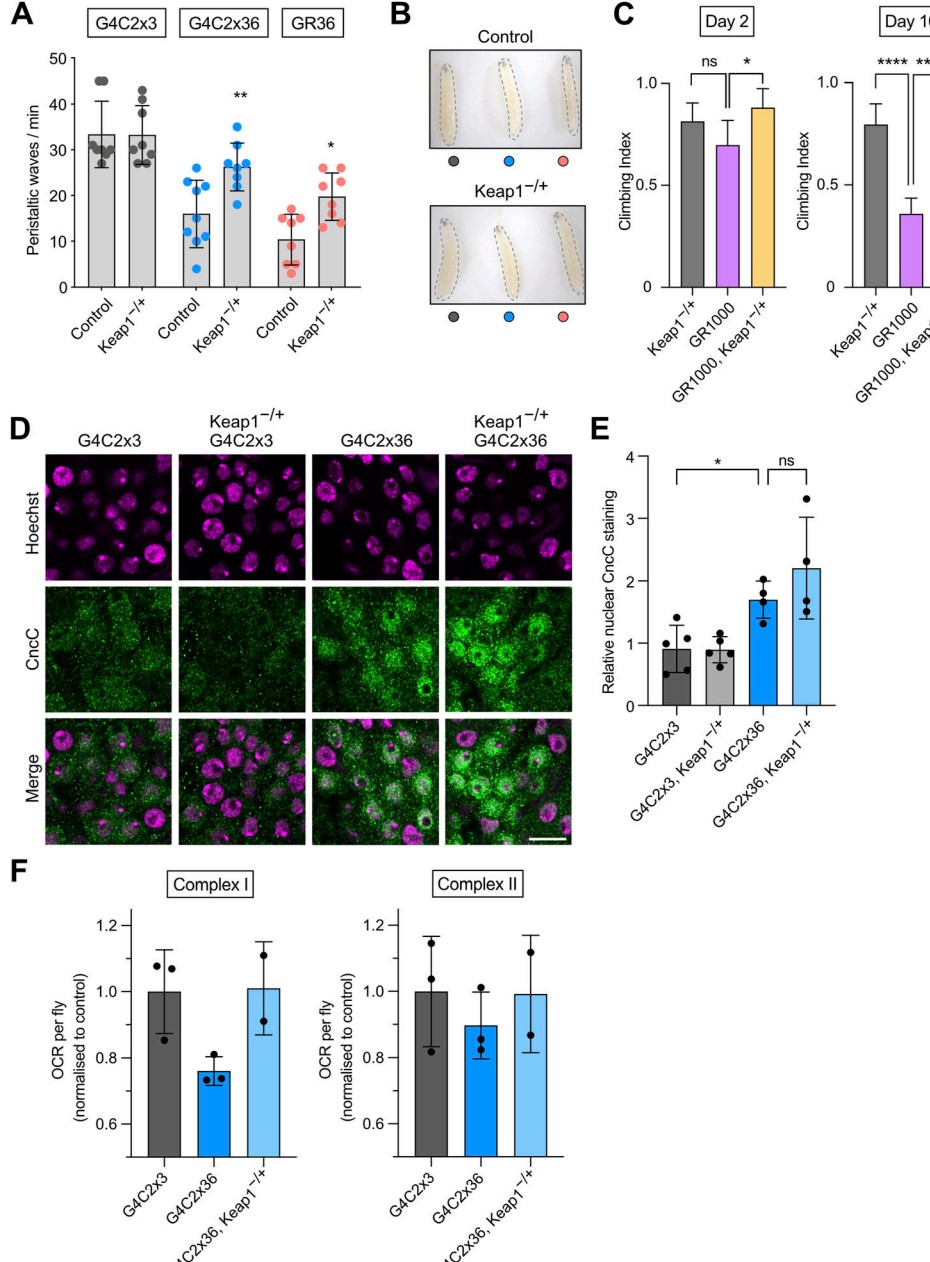

**Figure 5. Genetic activation of the Keap1/CncC pathway rescues *C9orf72* toxicity in vivo.**
**(A, B)** Locomotor ability (crawling) and (B) morphology of larvae pan-neuronally expressing G4C2x3, G4C2x36, or GR36 larvae using *nSyb*-GAL4, with heterozygous loss of *Keap1* (*Keap1⁻/⁺*) compared with a control (WT) background (mean ± SD; one-way ANOVA with Bonferroni's multiple comparison test, *P < 0.05, **P < 0.01; n = 8–10 larvae). **(C)** Climbing analysis of 2- and 10-d-old adults pan-neuronally (*nSyb*-GAL4) expressing GR1000 with heterozygous loss of *Keap1* (mean ± 95% CI; Kruskal–Wallis non-parametric test with Dunn's correction, *P < 0.05, ****P < 0.0001, ns = non-significant; n = 60–100 flies). **(D)** Confocal microscopy of larval ventral ganglion immunostained for CncC expression, with Hoechst used to delineate the nuclei, in larvae expressing pan-neuronal G4C2x3 or G4C2x36, via *nSyb*-GAL4, with or without heterozygous loss of *Keap1*; scale bar = 10 *µm*. **(E)** Quantification of relative nuclear CncC staining normalised to the control, G4C2x3 (mean ± SD; one-way ANOVA with Bonferroni's multiple comparison test for larval brain analysis, ns = non-significant, *P < 0.05; n = 4–5 larval brains). **(F)** Oxygen consumption rate of 5-d-old fly heads of G4C2x3 or G4C2x36, via *nSyb*-GAL4, and G4C2x36, *Keap1⁻/⁺* (mean ± SD; n = 2–3 biological replicates).

also significantly reduced with 20 d of DMF treatment, compared with the vehicle control (Fig 6D). Taken together, these data indicate that inhibiting Keap1 by DMF treatment can induce an antioxidant response transcriptional profile, alleviate mitochondrial dysfunction, and attenuate *C9orf72* behavioural phenotypes.

### DMF treatment is beneficial in *C9orf72*-ALS/FTD patient-derived neurons

Since we had observed elevated ROS in *C9orf72 Drosophila* models and found that DMF treatment provided phenotypic benefit, we wanted to test the translatable potential of DMF treatment on patient-relevant material. Fibroblasts from ALS patients carrying *C9orf72* mutations and healthy age–matched controls (Table 2), previously reported and successfully differentiated (42), were reprogrammed into induced neural progenitor cells (iNPCs), then differentiated into induced neurons (iNeurons), as assessed by the expression of neuronal markers *β*III-tubulin, MAP2, and NeuN (Fig S3A and B). Although these cultures did not show overt signs of excess toxicity or cell death (Fig S3C), assessing the level of oxidative stress using MitoSOX, we found a modest increase in mitochondrial ROS in *C9orf72* iNeurons (Fig 7A and B), in line with that observed in *Drosophila*. We also found that there was a higher level of nuclear Nrf2 in patient iNeurons (Fig 7C and D) mirroring the *C9orf72 Drosophila* phenotypes.

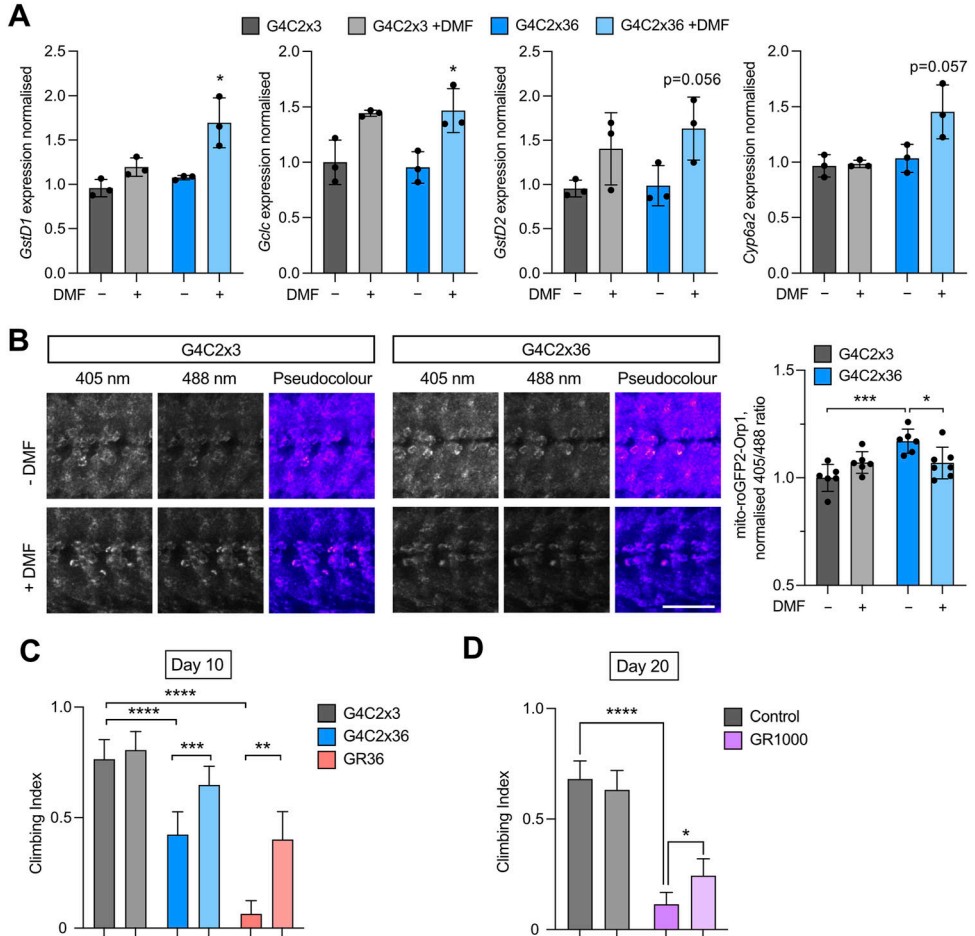

**Figure 6. Treatment with DMF rescues *C9orf72* toxicity in vivo.**
**(A)** Expression analysis (qRT–PCR) performed on 5-d-old adult heads pan-neuronally expressing G4C2x3 or G4C2x36 via *nSyb*-GAL4, with or without feeding DMF (7 μM), analysing mRNA levels of *GstD1*, *Gclc*, *GstD2*, and *Cyp6a2* (mean ± SD; unpaired *t* test with Welch's correction, *P < 0.05; n = 3 biological replicates). **(B)** Confocal microscopy of the mito-roGFP2-Orp1 mitochondrial $H_2O_2$ reporter co-expressed with G4C2x3 or G4C2x36 in the larval ventral ganglion via *nSyb*-GAL4. Larvae were fed 1 μM DMF or equivalent ethanol control during development. Representative pseudocolour ratio images are shown, and the 405/488 ratio is quantified. Scale bar = 50 μm (mean ± SD; one-way ANOVA with Bonferroni's multiple comparison test, *P < 0.05, ***P < 0.001; n = 6–7 larvae). **(C)** Climbing analysis of 10-d-old flies expressing G4C2x3, G4C2x36, or GR36 flies via a predominantly motor neuron driver *DIPγ*-GAL4. Flies were raised on food supplemented with either 7 μM dimethyl fumarate (DMF) or ethanol as a control (mean ± 95% CI; Kruskal–Wallis non-parametric test with Dunn's correction, **P < 0.01, ***P < 0.001, ****P < 0.0001; n = 60–100 flies). **(D)** Climbing analysis of 20-d-old flies expressing mito.GFP (control) or GR1000 flies via pan-neuronal expression with *nSyb*-GAL4. Flies were raised on food supplemented with either 7 μM DMF or ethanol as a control (mean ± 95% CI; Kruskal–Wallis non-parametric test with Dunn's correction, *P < 0.05, ****P < 0.0001, ns = non-significant; n = 60–100 flies).

**Table 2. Cell lines used in this study.**

| Cell line | Genotype | Sex | Age at biopsy (years) | Origin |
|---|---|---|---|---|
| 155 | Control | Male | 40 | University of Sheffield |
| 3050 | Control | Male | 68 | University of Sheffield |
| CS14 | Control | Female | 52 | Cedars–Sinai |
| 183 | *C9orf72* | Male | 50 | University of Sheffield |
| 78 | *C9orf72* | Male | 66 | University of Sheffield |
| ALS52 | *C9orf72* | Male | 49 | Cedars–Sinai |

Testing two different concentrations, 24-h treatment with 3 or 10 μM DMF was able to reduce the mitochondrial ROS levels in *C9orf72* iNeurons (Fig 7A and B) without detrimentally affecting viability (Fig S3D). By comparison, we analysed treatment with edaravone (EDV), a known ROS scavenger and an FDA-approved ALS drug, as a positive control, and observed that 24-h treatment with 30 and 100 μM EDV reduced ROS levels in *C9orf72* iNeurons to a similar extent as DMF (Fig S3E–G). Although significant, the effect size of DMF or EDV treatment is small and the impact on neuronal function or survival is currently unknown. Hence, more investigation is required to validate these initial results.

Nevertheless, taken together, these data suggest that activating the Keap1/Nrf2 pathway using therapeutic interventions, such as DMF, can be beneficial across multiple model systems including in vivo *C9orf72 Drosophila* models and human *C9orf72* patient-derived neurons.

## Discussion

We have conducted a comprehensive characterisation of mitochondrial dysfunction using three different models of *C9orf72* ALS/FTD and

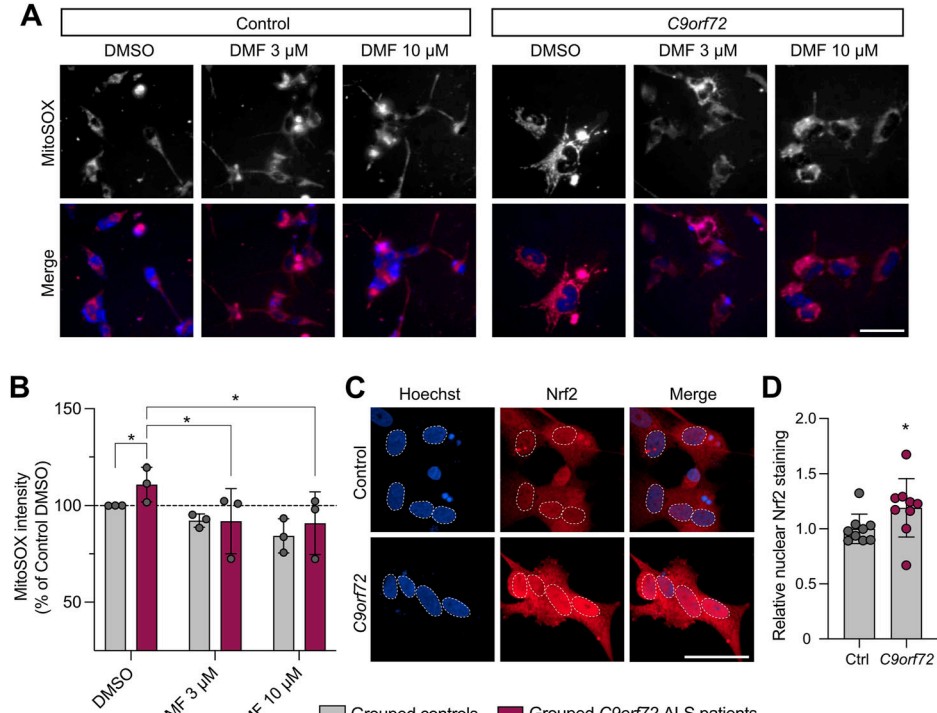

**Figure 7. DMF treatment reduces mitochondrial ROS in *C9orf72* patient-derived iNeurons.**
**(A)** MitoSOX staining of *C9orf72* iNeurons compared with healthy controls treated with 3 μM and 10 μM DMF or DMSO as a control. Scale bar = 50 μm. **(B)** Quantification of MitoSOX staining (mean ± SD; two-way ANOVA with Dunnett's multiple comparison test, *$P < 0.05$; n = 3 age-matched patients versus control). **(C)** Confocal microscopy images of *C9orf72* iNeurons compared with healthy controls immunostained for Nrf2 expression, with Hoechst used to identify the nuclei; scale bar = 50 μm. **(D)** Quantification of relative nuclear staining (mean ± SD; unpaired *t* test with Welch's correction, *$P < 0.05$; n = 3 age-matched patients versus control).

found disrupted morphology with hyperfused mitochondria, reduced mitophagy, impaired respiration, and increased mitochondrial ROS production, all in a neuronal context in vivo. Genetic interaction studies showed that only the overexpression of mitochondrial *Sod2* and *catalase* was able to significantly rescue *C9orf72* behavioural phenotypes, and further rescued other mitochondrial deficits such as mitophagy. Together, these data suggest a causative link between mitochondrial dysfunction, ROS and behavioural phenotypes.

Perturbed ROS levels have been well characterised in the context of ALS (25); however, it is important to understand the relative contributions of different ROS species in different cellular compartments in order to identify more targeted treatment options. By addressing this, we have observed a robust increase in mitochondrial ROS, whereas the effect on cytosolic ROS was more variable or unaltered. Consistent with the compartmentalised effects, we found that overexpressing mitochondrial *Sod2* or a mitochondrially targeted *Cat* partially rescued *C9orf72* phenotypes, as did the overexpression of cytosolic *Cat*. This suggests that mitochondrial superoxide produced is rapidly dismutated causing a cytosolic effect; hence, both cytosolic and mitochondrially targeted *Cat* are beneficial. We were surprised to find that *Sod1* overexpression exacerbated *C9orf72* phenotypes, contrary to previous findings from Lopez-Gonzalez et al (20). SOD1 is a cytosolic ubiquitous enzyme with several functions, primarily involved in scavenging superoxide, as well as modulating cellular respiration, energy metabolism, and post-translational modifications (43). Mutations in *SOD1* also cause ALS, most likely via misfolding of the protein, causing a toxic gain of function. However, the exact mechanism leading to motor neuron death is still elusive (44). Together, these findings highlight the importance of studying

compartmentalised ROS effects in the context of disease pathology.

While defective autophagy has previously been noted in *C9orf72 Drosophila* and other models (45), we extend these observations to provide in vivo evidence that mitophagy is also perturbed. Currently, a mechanistic link between disrupted mitophagy and the upstream oxidative stress is lacking. However, Nrf2 activity is known to promote mitochondrial homeostasis at multiple levels including biogenesis and turnover (46). Notably, the autophagy adapter p62 both competes with Nrf2 for Keap1 binding (47) and is an Nrf2 target gene, thereby creating a positive feedback loop (47, 48). Moreover, Nrf2 activation was shown to induce mitophagy independently of the Pink1/parkin pathway, and rescue *Pink1/parkin* phenotypes in vivo (49, 50). Therefore, it will be interesting to investigate a potential link between *C9orf72*-Nrf2/CncC-p62/ref(2)P-mitophagy/autophagy as it may provide mechanistic insight, in all of our model systems.

The protective role of the antioxidant and cytoprotective Keap1/Nrf2 pathway has been discussed as a therapeutic target for treatment against many neurodegenerative diseases including ALS (24). Investigating this, we found increased nuclear localisation of Nrf2 and concomitant up-regulation of the *GstD1-GFP* reporter, a proxy for Nrf2 activation, indicating that the pathway is up-regulated upon *C9orf72* toxicity. However, despite the initiated activation, this was clearly insufficient to prevent excessive ROS and the resulting neurotoxicity, which could be prevented by the transgenic overexpression of antioxidant enzymes. From a translational perspective, it is encouraging that both genetic reduction of Keap1 and its pharmacological targeting with DMF could suppress *C9orf72* toxicity in vivo, and in

patient-derived iNeurons. Testing alongside the approved ALS drug EDV, it was noteworthy that the effects of DMF on reducing ROS were comparable to EDV but at an order of magnitude lower concentration. It will be interesting for future work to further evaluate the mechanistic benefits versus detrimental side effects of different concentrations of DMF and EDV. Regarding the analysis of patient-derived iNeurons, it will be important for future work to assess the mitochondrial/oxidative stress phenotypes and suppression by DMF and/or EDV in iNeurons compared with isogenic controls.

While some studies have reported the Nrf2 pathway to be disrupted in ALS patient samples (51, 52, 53), very limited data exist on C9orf72-related ALS samples. Recently, Jimenez-Villegas et al (54) have shown that Nrf2 activation is impaired in cell culture models of arginine DPR toxicity, where they also saw an improvement in cell viability upon treatment with DMF. We have extended these findings showing the benefit of targeting Keap1/Nrf2 in vivo and in patient-derived iNeurons, which further emphasises the importance of Nrf2 activation as a potential therapeutic target. Recently, a phase 2 clinical trial to test the efficacy of DMF in sporadic ALS patients was conducted, which presented some encouraging results (55). Although the study concluded there was no significant effect on the primary endpoint (ALS Functional Rating Scale-Revised score), there was a reduced decline in the neurophysiological index, suggesting preservation of lower motor neuron function. The authors also noted that participants showed an "unusually slow disease progression" and that a larger trial is needed for verification.

In conclusion, our results provide compelling evidence that mitochondrial oxidative stress is an important proximal pathogenic mechanism leading to downstream mitochondrial dysfunction such as alterations in mitochondrial function and turnover. Consequently, targeting one of the main intracellular defence mechanisms to counteract oxidative stress—the Keap1/Nrf2 signalling pathway—could be a viable therapeutic strategy for ALS/FTD. Although DMF treatment shows promise, more research is needed to understand the underlying mechanisms behind disease pathogenesis and progression.

# Materials and Methods

## Drosophila stocks and husbandry

Flies were raised under standard conditions in a temperature-controlled 12:12-h light/dark cycle incubator at 25°C and 65% relative humidity, on standard Drosophila food containing cornmeal, agar, molasses, yeast, and propionic acid (Genetics Department, University of Cambridge). Transgene expression was induced using the pan-neuronal driver nSyb-GAL4 or the predominantly motor neuron driver DIPγ-GAL4. The following strains were obtained from Bloomington Drosophila Stock Center: w[1118] (BDSC_6326), nSyb-GAL4 (BDSC_51635), UAS-mito-HA-GFP (BDSC_8443), UAS-luciferase RNAi (BDSC_31603), UAS-Sod1 (BDSC_33605), UAS-Sod2 (BDSC_24492), UAS-catalase (BDSC_24621), Opa1[s3475] (BDSC_12188), UAS-GFP.mCherry.Atg8a (BDSC_37749), UAS-mito-roGFP2-Orp1 (BDSC_67667), UAS-cyto-Grx1-roGFP2 (BDSC_67662), UAS-mito-roGFP2-Grx1 (BDSC_67664); the Zurich ORFeome Project: UAS-LacZ (F005035) and UAS-Tango11.HA (F002828); and the Fly Stocks of National Institute of Genetics: UAS-USP30 RNAi (3016R-2). Other Drosophila lines were kindly provided as follows: DIPγ (MI03222)-GAL4 (BDSC_90315) from Dr Robert Carrillo (56, 57), UAS-G4C2x3, UAS-G4C2x36, and UAS-GR36 from Prof. Adrian Isaacs (6), UAS-GR1000-eGFP from Dr Ryan West (32), UAS-mito.Catalase from Prof. Alberto Sanz (58), UAS-Drp1.WT from Prof. Jongkyeong Chung (59), Marf[B] from Prof. Hugo Bellen (60), and GstD1-GFP (36) and Keap1[del] (61) from Prof. Linda Partridge. The UAS-mito-QC reporter (BDSC_91640) has been described before (30).

## Drug treatments

For paraquat treatment, standard Drosophila food was supplemented with paraquat (856177; Sigma-Aldrich) to a final concentration of 10 mM. For dimethyl fumarate (DMF, 242926; Sigma-Aldrich) adult treatment, DMF (or an equivalent volume of ethanol for the vehicle control) was added to a sugar–yeast (SY) medium (61) consisting of 15 g/litre agar, 50 g/litre sugar, and 100 g/litre yeast to a final concentration of 7 μM. All flies were transferred into freshly prepared supplemented food every 2–3 d. For larval DMF treatment, crosses of the parental genotypes were set up in standard food containing 1 μM DMF (or an equivalent volume of ethanol for the vehicle control), providing the larval offspring exposure to the treatment throughout development.

## Locomotor and survival assays

### Larval crawling
Larval crawling was conducted using wandering third-instar (L3) larvae. Each larva was placed in the middle of a 1% agar plate, where they were left to acclimatise for 30 s. After, the number of forward and backward peristaltic waves was counted for 60 s and recorded.

### Climbing
The repetitive iteration startle-induced negative geotaxis (RISING or "climbing") assay was performed using a counter-current apparatus as previously described (62). Briefly, groups of 15–22 flies were placed in a temperature-controlled room for 30 min for temperature acclimatisation and transferred to test tubes for another 30 min. Flies were placed into the first chamber, tapped to the bottom, and given 10 s to climb a 10-cm distance. 20-d-old flies were given 20 s to climb to account for overall reduced mobility. Flies that reached the upper portion, that is, climbed 10 cm or more, were shifted into the adjacent chamber. After five successful trials, the number of flies in each chamber was counted and the average score was calculated and expressed as a climbing index.

### Survival
For survival lifespan experiments, groups of 20 males each were collected with minimal time (<30 s) under light anaesthesia and placed in separate food vials in standard food and transferred every 2–3 d to fresh food, and the number of dead flies was recorded. Per cent survival was calculated at the end of the experiment using https://flies.shinyapps.io/Rflies/ (Luis Gracia).

### Oxidative stress assays

For short-term hydrogen peroxide ($H_2O_2$) and paraquat (PQ) assays, flies were kept in vials containing filter paper soaked in 5% wt/vol sucrose solution containing either 1% $H_2O_2$ (10386643; Thermo Fisher Scientific) or 10 mM PQ (856177; Sigma-Aldrich). Deaths were recorded two times per day, and flies were transferred every 2–3 d to fresh tubes.

### Immunohistochemistry and sample preparation

For mitophagy analysis with the mito-QC reporter, autophagy analysis with the GFP.mCherry.Atg8a reporter, and mitochondrial morphology analysis, third-instar larval brains were dissected in PBS and fixed in 4% formaldehyde (FA) (28908; Thermo Fisher Scientific) in PBS for 20 min at RT. For the mitophagy and autophagy experiments, the 4% FA/PBS was adjusted to pH 7. Samples were then washed in PBS followed by water to remove salts. ProLong Diamond Antifade mounting medium (P36961; Invitrogen) was used to mount the samples and imaged the next day.

For immunostaining of third-instar larval brains, larvae were dissected and fixed as described, permeabilised in 0.3% Triton X-100 in PBS (PBS-T) for 30 min, and blocked with 1% BSA in PBS-T for 1 h at RT. Tissues were then incubated with mouse anti-CncC (63) (1:500; a kind gift from Dr Fengwei Yu), diluted in 1% BSA in PBS-T overnight at 4°C, then washed three times for 10 min with PBS-T, followed by incubation with secondary antibody—goat anti-mouse IgG H&L Alexa Fluor 488 (1:500, A11001; Invitrogen). The tissues were washed in PBS-T with 1:10,000 Hoechst (H3570; Invitrogen) for 20 min followed by two times of PBS washes and mounted on slides using ProLong Diamond Antifade mounting medium.

For immunostaining of adult brains, flies were dissected in PBS and fixed on ice in 4% FA in PBS for 30 min. Brains were washed three times for 20 min in PBS-T, before blocking for 4 h in 4% normal goat serum in PBS-T. Tissues were then incubated with mouse anti-CncC (63) (1:500; a kind gift from Dr Fengwei Yu), diluted in 4% normal goat serum in PBS-T overnight at 4°C, then washed three times for 10 min with PBS-T, followed by incubation with secondary antibody for 2 h. The tissues were washed in PBS-T with 1:10,000 Hoechst for 20 min followed by two times of PBS washes and mounted on slides using Prolong Diamond Antifade mounting medium.

### ROS analysis

#### MitoSOX

10-d-old adult brains were dissected in PBS, incubated in 20 $\mu$M of MitoSOX Red (M36008; Invitrogen) for 30 min in the dark, washed with PBS for three times, mounted on poly-L-lysine–coated wells on a 1.5-mm coverslip, and imaged live immediately. The maximum intensity of projected z-stacks from imaged brains was quantified using ImageJ.

#### Dihydroethidium (DHE)

2–3 larval brains were dissected in HL3 (5 mM Hepes, 70 mM NaCl, 5 mM KCl, 20 mM MgCl$_2$, 1.5 mM CaCl$_2$, 5 mM trehalose, 115 mM sucrose) (64) and incubated for 15 min in the dark at RT in 30 $\mu$M of DHE (D11347;

Invitrogen). Samples were then washed in HL3 for 10 min before imaging live in a drop of HL3 on poly-L-lysine–coated wells on 1.5-mm coverslips.

### Genetic roGFP2 reporters

Mitochondrial and cytosolic ROS imaging was performed using the *mito-roGFP2-Orp1, mito-roGFP2-Grx1*, and *cyto-Grx1-roGFP2* reporter lines. Third-instar larval brains were dissected in HL3, placed in a drop of HL3 on poly-L-lysine–coated wells on a 1.5-mm coverslip, and imaged by excitation at 488 nm (reduced) or 405 nm (oxidised), with emission detected at 500–530 nm. The maximum intensity of projected z-stacks from imaged brains was quantified using FIJI (ImageJ), and the ratio of 405/488 nm was calculated.

### Microscopy

Fluorescence imaging was conducted using a Zeiss LSM 880 confocal microscope (Carl Zeiss MicroImaging) equipped with Nikon Plan-Apochromat 63x/1.4 NA oil immersion objectives. Images were prepared using FIJI (ImageJ). For mito-QC imaging, the Andor Dragonfly spinning disc microscope was used, equipped with a Nikon Plan-Apochromat 100x/1.45 NA oil immersion objective and an iXon camera. Z-stacks were acquired with 0.2-$\mu$m steps. For larval morphology, images were acquired using a Leica DFC490 camera mounted on a Leica MZ6 stereomicroscope.

### Quantification and analysis methods

#### Mitophagy

A single-image stack was generated per animal, in which at least 10 cells were analysed. Confocal images were processed using FIJI (ImageJ). The quantification of mitolysosomes was performed as described in reference (30) using Imaris (version 9.0.2) analysis software. Briefly, a rendered 3D surface was generated corresponding to the mitochondrial network (GFP only). This surface was subtracted from the mCherry signal, which overlapped with the GFP-labelled mitochondrial network, defining the red-only mitolysosome puncta with an estimated size of 0.5 $\mu$m and a minimum size cut-off of 0.2 $\mu$m diameter determined by Imaris.

#### Autophagy

The quantification of autolysosomes was performed using FIJI (ImageJ) with the 3D Objects Counter plugin. An area of interest was selected by choosing 6–10 cells per image. The threshold was based on matching the mask with the fluorescence. A minimum size threshold of 0.05 $\mu$m$^3$ was set to select autolysosomes.

#### Mitochondrial morphology

After acquisition of images, each cell was classified using a scoring system where morphology was scored as fragmented, WT/tubular, or fused/hyperfused. All images were blinded and quantified by three independent investigators. Data presented in Figs 3 and S2 were conducted concurrently; therefore, the control groups are the same but replicated in the two figures for ease of reference.

### Nuclear CncC quantification

All acquired images were taken with the same laser and gain during acquisition, which allowed a threshold to be set in FIJI (ImageJ) that was consistent for all images. For each brain, using the Hoechst signal, 10 nuclei from the central part of the larval CNS and 10 from the periphery were quantified to minimise bias. This was overlaid onto the CncC channel, and the mean intensity within the nuclei was measured using the ROI manager. All images were blinded before quantification.

### Mitochondrial respiration

Mitochondrial respiration was monitored at 25°C using an Oxygraph-2k high-resolution respirometer (OROBOROS Instruments). Standard oxygen calibration was performed before the start of every experiment. 25-d-old adult fly heads per replicate for each genotype were extracted using forceps and placed in 100 $\mu$l of respiration buffer (RB) (120 mM sucrose, 50 mM KCl, 20 mM Tris–HCl, 4 mM KH$_2$PO4, 2 mM MgCl$_2$, 1 mM EGTA, and 1 g/litre fatty acid–free BSA, pH 7.2). This was homogenised on ice using a pestle with 20 strokes. 1 ml of RB was added to the homogenate and passed through a 1-ml syringe with a piece of cotton wool inside to remove the debris. This was repeated with another 1 ml of RB. In total, 2.1 ml of homogenate was added to the respiratory chambers. For coupled "state 3" assays, saturating concentrations of substrates including 10 mM glutamate, 2 mM malate, 10 mM proline, and 2.5 mM ADP were added to measure complex I–linked respiration. 0.15 $\mu$M rotenone was added to inhibit complex I, and 10 mM succinate was added to measure complex II–linked respiration. Data acquisition and analysis were carried out using DatLab software (OROBOROS Instruments).

### Immunoblotting

Proteins were isolated from either larval brains or adult heads using RIPA lysis buffer (50 mM [pH 7.4] Tris, 1 M NaCl, 0.1% SDS, 0.5% sodium deoxycholate, and 1% NP-40) supplemented with cOmplete mini EDTA-free protease inhibitors (Roche). After protein quantification using the bicinchoninic acid assay (23225; Thermo Fisher Scientific), Laemmli buffer (1610747; Bio-Rad) containing 1:10 $\beta$-mercaptoethanol (M6250; Sigma-Aldrich) was added. Samples were boiled at 95°C for 10 min, resolved by SDS–PAGE using 4–20% or 10% precast gels (4561093; Bio-Rad), depending on the molecular weight of the desired protein, and transferred onto a nitrocellulose membrane (1704158; Bio-Rad) using a Bio-Rad Trans-Blot semi-dry system. Membranes were blocked with 5% (wt/vol) dried skimmed milk powder (Marvel Instant Milk) in TBS with 0.1% Tween-20 (TBS-T) for 1 h at RT and probed with the appropriate primary antibodies diluted in TBS-T overnight at 4°C. After three 10-min washes in TBS-T, the membranes were incubated with the appropriate HRP-conjugated secondary antibodies diluted in 5% milk in TBS-T for 1 h at RT. Membranes were washed three times for 10 min in TBS-T, and detection was achieved with the Amersham ECL Prime detection kit (RPN2232; Cytiva). Blots were imaged using the Amersham Imager 680 with further analysis and processing using FIJI (ImageJ).

The following primary antibodies for immunoblotting were used in this study: GFP (1:100, ab294; Abcam), tubulin (1:5,000, T9026; Sigma-Aldrich), ref(2)P (1:1,000, ab178440; Abcam), and GABARAP (1:1,000, ab109364; Abcam). The following secondary antibodies for immunoblotting were used in this study: goat anti-mouse IgG H&L (HRP-conjugated, 1:10,000, ab6789; Abcam) and goat anti-rabbit IgG H&L HRP (1:10,000, G21234; Thermo Fisher Scientific).

### qRT–PCR

Quantitative real-time PCR (qRT–PCR) was carried out as follows: 30 heads were collected and placed in a 2-ml tube containing 1.4-mm ceramic beads (15555799; Fisherbrand) for tissue preparation. 400 $\mu$l of TRI Reagent (T9424; Sigma-Aldrich) was added and placed into Minilys homogeniser (Bertin Technologies) where the programme was set to a maximum speed of 10 s. The samples were placed back on ice for 5 min before two further rounds of lysis. The Direct-zol RNA MiniPrep kit (R2050; Zymo Research) was used to extract RNA following the manufacturer's instructions. TURBO DNA-free Kit (AM1907; Invitrogen) was used to remove contaminating DNA following the manufacturer's instructions. cDNA synthesis was achieved using Maxima H Minus cDNA Synthesis Kit (M1681; Thermo Fisher Scientific) following the manufacturer's instructions. Equivalent (500 ng) total RNA underwent reverse transcription for each sample. Finally, qRT–PCR was run using Maxima SYBR Green/ROX Kit (K0221; Thermo Fisher Scientific) following the manufacturer's instructions using the QuantStudio 3 RT–PCR machine. The relative transcript levels of each target gene were normalised to a geometric mean of *RpL32* and *Tub84b* reference genes; relative quantification was performed using the comparative $C_T$ method entering into account PCR primer efficiency (65).

The following primers were used in this study:
*Sod1*: CCAAGGGCACGGTTTTCTTC, CCTCACCGGAGACCTTCAC.
*Sod2*: GTGGCCCGTAAAATTTCGCAAA, GCTTCGGTAGGGTGTGCTT.
*Catalase*: CCAAGGGAGCTGGTGCTT, ACGCCATCCTCAGTGTAGAA.
*GstD1*: CCGTGGGCGTCGAGCTGAACA, GCGCGAATCCGTTGTCCACCA.
*GstD2*: AACCAGCGTCTGTACTTCGA, TCAAGTCCTCATCCGATCCG.
*Cyp6a2*: AAGCCATGACCTACTTGAACC, TGCCCTTCTCAATCACAAGC.
*Gclc*: TCCTCTCAGTTCAGCCCG, TTCGTCTTTGTGTCCTTGAAAAC.
*RpL32*: AAACGCGGTTCTGCATGAG, GCCGCTTCAAGGGACAGTATCTG.
*Tub84b*: TGGGCCCGTCTGGACCACAA, TCGCCGTCACCGGAGTCCAT.

### iNPC tissue culture, iNeuron differentiation, and compound treatment

Induced neuronal progenitor cells (iNPCs) were generated as previously described (66, 67). Details for the patient lines used in this study are provided in Table 2. iNPCs were maintained in DMEM/F-12 with GlutaMAX (31331028; Gibco) supplemented with 1% N-2 (17502001; Invitrogen), 1% B-27 (17504001; Invitrogen), and 20 ng/ml FGF-basic (100-18B; PeproTech) on fibronectin (FC010; Millipore)-coated cell culture dishes. Cells were routinely subcultured every 2–3 d using Accutase (15323609; Corning) to detach them. To achieve neuronal differentiation, iNPCs were plated into six-well plates and cultured for 2 d in DMEM/F-12 media with GlutaMAX supplemented with 1% N-2, 2% B-27, and 0.5 $\mu$M DAPT (D5942; Sigma-Aldrich). On day 3, DAPT was removed, and media were supplemented with 0.5 $\mu$M smoothened agonist (SAG, 566660; Millipore), 2.5 $\mu$M forskolin

(11018; Cayman Chemical), and 1 $\mu$M retinoic acid (R2625; Sigma-Aldrich) for 16 d Cells were treated with edaravone (EDV; M70800; Sigma-Aldrich) and DMF (242926; Sigma-Aldrich) for 24 h before imaging. Cells were treated with 30 or 100 $\mu$M EDV, or DMF at concentrations of 3 or 10 $\mu$M.

### Live-cell imaging assays of iNeurons

Cells were stained for 30 min with Hoechst (B2883; Sigma-Aldrich) and MitoSOX Red (M36008; Invitrogen) at concentrations of 20 $\mu$M and 500 nM, respectively. Cells were imaged using an Opera Phenix high-content imaging system, with analysis performed using a custom protocol on Harmony software (PerkinElmer). Nucleus counts after treatments with DMF and EDV were normalised to baseline counts per cell line. Cell viability was measured using a viability indicator from Neurite Outgrowth Staining Kit (A15001; Thermo Fisher Scientific) according to the manufacturer's instructions.

### Fixed cell imaging of iNeurons

On day 18 of differentiation, cells were fixed in 4% PFA for 30 min. After PBS washes, cells were permeabilised with 0.1% Triton X-100 for 10 min and blocked with 5% horse serum (H0146; Sigma-Aldrich) for 1 h. Cells were incubated with primary NRF2 antibody overnight at 4°C. Cells were washed with 0.1% Tween in PBS and incubated with secondary antibodies and 1 $\mu$M Hoechst before imaging. Cells were imaged using an Opera Phenix high-content imaging system, with analysis performed using a custom protocol on Harmony software (PerkinElmer).

The following primary antibodies for immunostaining of iNeurons were used in this study: NRF2 (1:1,000, ab31163; Abcam), $\beta$III-tubulin (1:1,000, AB9354; Merck), MAP2 (1:1,000, ab32454; Abcam), NeuN (1:1,000, ab177487; Abcam). The following secondary antibodies for immunostaining of iNeurons were used in this study: donkey anti-rabbit IgG H&L Alexa Fluor 568 secondary antibody (1:1,000, A10042; Invitrogen) and goat anti-chicken IgG H&L Alexa Fluor 488 secondary antibody (1:1,000, A11039; Invitrogen).

### Statistical analysis

GraphPad Prism 9 (SCR_002798; RRID) was used to perform all statistical analyses. Climbing data were analysed using the Kruskal–Wallis non-parametric test with Dunn's correction for multiple comparisons. Data are presented as the mean ± 95% confidence interval (CI). Quantifications of larval crawling, the number of mitolysosomes and autolysosomes, WB, and qRT–PCR were analysed using one-way ANOVA with the Bonferroni post hoc test for multiple comparisons. Data are presented as the mean ± SD. Mitochondrial morphology was quantified using a chi-squared test. Unpaired t tests with Welch's correction for unequal SD were used for respiratory analysis and immunoblots. Two-way ANOVA with Dunnett's multiple comparison test was used for analysis of the iNeurons work. Lifespan experiments were analysed using the Mantel–Cox log-rank test. All statistical tests and n numbers are stated in the figure legends.

## Data Availability

All data needed to evaluate the conclusions in the study are present in the study and/or the Supplementary Materials. This study does not include data deposited in external repositories.

### Ethics statement

For the iNeuron work, informed consent was obtained from all participants before collection of fibroblast biopsies (Sheffield Teaching Hospital study number: STH16573, Research Ethics Committee reference: 12/YH/0330); fibroblasts obtained by Cedars–Sinai are covered by MTA. *Drosophila* stocks were obtained from the Bloomington *Drosophila* Stock Center, which is supported by grant NIH P40OD018537. Experimental work with *Drosophila* is exempt from ethical approval as they are not covered by the Animals (Scientific Procedures) Act 1986.

## Supplementary Information

## Acknowledgements

We kindly thank Dr Robert Carrillo, Prof. Adrian Isaacs, Dr Ryan West, Prof. Alberto Sanz, and Prof. Linda Partridge for generously sharing fly lines, as well as Dr Fengwei Yu for sharing the CncC antibody. We also thank Sarah Granger and Katie Roome for technical assistance with the iNeuron work. Finally, we thank all members of the Whitworth laboratory for discussions during the project and feedback on the article. This work was supported by the Medical Research Council (MRC) core funding (MC_UU_00015/6 and MC_UU_0028/6) to AJ Whitworth, MRC studentships (MC_ST_U18009) to WH Au and MJ Twyning, and MRC project grants MR/V003933/1 to AJ Whitworth/L Miller-Fleming and MR/W00416X/1 to L Ferraiuolo. JAK Lee was supported by Battelle Memorial Institute Wadsworth PhD Fellowship. H Mortiboys/JAK Lee were supported by MNDA 943-793, and PJ Shaw was supported by the NIHR Sheffield Biomedical Research Centre (NIHR 203321). H Mortiboys was supported by Parkinson's UK Senior Research Fellowship F-1301. This is independent research funded by the above funders and partially carried out at the National Institute for Health and Care Research (NIHR) Sheffield Biomedical Research Centre (BRC). The views expressed are those of the author(s) and not necessarily those of the funders, the NIHR, or the Department of Health and Social Care.

### Author Contributions

WH Au: conceptualisation, formal analysis, investigation, visualisation, methodology, and writing—original draft, review, and editing.

L Miller-Fleming: conceptualisation, investigation, methodology, and writing—review and editing.

A Sanchez-Martinez: supervision, methodology, and writing—review and editing.

JAK Lee: formal analysis, investigation, methodology, and writing—review and editing.

MJ Twyning: investigation, methodology, and writing—review and editing.

HA Prag: conceptualisation, methodology, and writing—review and editing.

L Raik: methodology and writing—review and editing.

SP Allen: resources, supervision, funding acquisition, and writing—review and editing.

PJ Shaw: resources, supervision, funding acquisition, and writing—review and editing.

L Ferraiuolo: supervision, methodology, and writing—review and editing.

H Mortiboys: conceptualisation, formal analysis, supervision, funding acquisition, and writing—review and editing.

AJ Whitworth: conceptualisation, formal analysis, supervision, funding acquisition, validation, project administration, and writing—original draft, review, and editing.

## Conflict of Interest Statement

The authors declare that they have no conflict of interest.

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
