## [Reviewer comments · Life Science Alliance]

Life Science Alliance

Activation of the Keap1/Nrf2 pathway suppresses mitochondrial dysfunction, oxidative stress and motor phenotypes in C9orf72 ALS/FTD models

Wing Hei Au, Leonor Miller-Fleming, Alvaro Sanchez-Martinez, James Lee, Madeleine Twyning, Hiran Prag, Laura Raik, Scott Allen, Pamela Shaw, Laura Ferraiuolo, Heather Mortiboys, and Alexander Whitworth

DOI: <https://doi.org/10.26508/lsa.202402853>

Corresponding author(s): Alexander Whitworth, University of Cambridge

Review Timeline:

Submission Date:	2024-05-30
Editorial Decision:	2024-05-31
Revision Received:	2024-06-04
Accepted:	2024-06-04

Transaction Report:

Please note that the manuscript was reviewed at *Review Commons* and these reports were taken into account in the decision-making process at *Life Science Alliance*.

Authors' Response to Reviewers

1. General Statements [optional]

This section is optional. Insert here any general statements you wish to make about the goal of the study or about the reviews.

We thank the reviewers for their evaluation of our previous submission. We have considered the critiques and suggestions, responding to each point in detail below, and where possible we have provided additional data to address the reviewers' points. Overall, the reviewers' comments were supportive of the work, though noting some limitations. In light of these comments, we have also revised the strength of our interpretations accordingly.

This section is mandatory. Please insert a point-by-point reply describing the revisions that were already carried out and included in the transferred manuscript.

Reviewer #1

Evidence, reproducibility and clarity (Required):

Au et al. used two fly models to study how mitochondrial defects are implicated in C9ALS, the most common familial ALS type. They found that in these flies, mitochondrial, but not cytosolic, ROS is upregulated, accompanied by locomotion defects agreeing with previous publications. Consistent with these data, sod2, but not sod1, rescues the behavioral defects in these flies. Also, manipulating mitochondrial dynamics or mitophagy does not rescue these defects. Furthermore, the authors showed that the Nrf2 activity is upregulated, likely due to oxidative stress, and genetically or pharmacologically suppressing the Keap1 function, which activates Nrf2 and thereby its downstream antioxidative genes, suppresses behavior defects in these flies. This part is generally solid and convincing, with minor issues that need some revision. Finally, the authors showed that mitochondrial ROS and nuclear Nrf2 are both upregulated in C9 iPS neurons, both of which are suppressed by the Keap1 inhibitor DMF, or a known antioxidant. For this part, the data are convincing but insufficient to support a good translation of their fly data.

Major concerns:

1a. The authors really need a phenotypic readout for their iPS experiments, either cell death or some sort of toxicity, to support the translatability of their fly data.

- We agree and appreciate the value of having such as phenotypic readout for the iPSC experiments but, unfortunately, within the context of the current work we did not observe any clear phenotype of toxicity or diminished viability under basal, unchallenged conditions. To support this, we have added our analysis of cell viability at the time of imaging, shown in new Supplementary Figure 3C and mentioned in the text (line 620-621).

1b. The authors also need to test the toxicity of DMF in iPS neurons.

- As above, we found that treatment with DMF conferred no overt toxicity within the time-course of our experiments. These data are shown in new Supplementary Figure 3D and mentioned in the text (line 626-628).

2. The authors should use genetic ways, e.g., knocking down Keap1, to activate Nrf2 and test whether this suppresses ROS and neurodegeneration phenotype in iPS neurons, as they did in flies.

3. They need to better characterize the Nrf2 activity in iPS neurons (see Minor Concern #1).

- Regarding these two points, we agree that it would be interesting to further investigate the Keap1/Nrf2 pathway in these cells, but time, personnel and resource constraints preclude additional investigations on this occasion. It is important to note that the cell models were used specifically to validate that elevated mitochondrial oxidative stress and increased nuclear Nrf2 localisation also occurred in patient-derived neurons, and whether DMF treatment could reverse the oxidative stress. This was the extent to which the cell models were used in this instance and the current data are sufficient to support the conclusions made based on this. We regret that it was not possible to delve deeper into this at the current time but will be possible in future work.

Minor concerns:

1a. Fig 4A and B are hard to comprehend. Can the authors show images with more obvious differences?

- We have now revised these figure panels replacing with alternative images. We hope that the new images show more appreciable differences. We understand that the differences can sometimes be subtle which is why we rely on the quantification for unbiased interpretation.

1b. Also, Gst-D1 is the only Nrf2 downstream gene tested. Can the authors use RT-PCR to test multiple genes? These will strengthen the point that Nrf2 is activated. Similar things should be done in iPS neurons.

- Thanks for this suggestion. To complement the immunoblots of the genomic GstD1-GFP reporter, we have now performed qRT-PCR on flies treated with or without DMF for additional Keap1/Nrf2 pathway targets, including *GstD1*, *Gclc*, *GstD2* and *Cyp6a2*. These data show that the degree of transcriptional activation was variable between different targets, but DMF treatment caused a general upregulation of CncC targets in G4C2x36 flies (new Fig. 6A).

2. What about cytosolic ROS in C9 iPS neurons? Is it similar to the fly models?

- We agree that this would be interesting to analyse. Unfortunately, given time and resource constraints we did not have the capacity to also explore this out of curiosity. Again, the specific focus for the iPSC neuron work was to validate the mitochondrial ROS aspect and action of DMF.

3. Unless the authors confirm that mitochondrial dynamics or mitophagy are not contributing to neurodegeneration in iPS neurons, I wouldn't emphasize their related negative data in flies. Overall, the authors need to tone down their arguments if the findings are not verified in iPS or other mammalian models.

- On reflection, we agree that the iNeuron data was given an overly prominent status within the study and we have adjusted the text accordingly throughout, including removing a specific mention of this in the title. That said, we still consider that the negative results regarding the lack of rescue of organism-scale phenotypes (e.g., locomotion) by manipulating mitochondrial dynamics or mitophagy to be important indicators of the relative mechanistic contribution of these processes to the organism-scale pathology (most closely reflecting the clinical condition). As discussed above (major point 1a), within the context of the current work we did not observe any clear phenotype of toxicity or diminished viability in the patient iNeurons. Therefore, it is not readily possible to test the relative contribution of mitochondrial dynamics vs mitophagy vs ROS to the survival of these cells, so we have based our interpretations of this on the *in vivo* models. In summary, we have toned down our statements relating to and stemming from data arising from the iNeuron work but our interpretation of the negative results in flies remains the same.

4. Can the authors measure the activities of OXPHOS complexes and ATP synthase/complex V?

- The intention of this study was to explore mechanisms that could alleviate pathological phenotypes *in vivo*. We have characterised a wide-range of cellular defects relating to mitochondrial dysfunction including overall OXPHOS function by OCR. Analysing individual OXPHOS complexes from animal tissue is not a trivial

undertaking and, other than providing a little more granularity to the nature of the respiratory defect, we considered that this would be a distraction from the main focus of the study.

5a. Edavarone is one of the only two effective drugs for general ALS, and it's believed to work as an antioxidant. The authors should discuss it along with relating their findings to therapeutic development.

- A statement on Edaravone being an FDA-approved treatment for ALS and an antioxidant (ROS scavenger) were included in the text (lines 628-629). We have added further comment on this in the Discussion (lines 686-690). Since edaravone was used as a comparator in this study, and to maintain the focus on DMF, we prefer to not elaborate on this further in the discussion.

5b. Also, the discussion on SOD1 aggregation sounds somewhat farfetched. Plus, it's not directly related to the central message of this paper. I would remove it.

- Fair enough. We have removed these statements from the text.

Significance (Required):

C9orf72-mediated ALS is the most common familial ALS type and also accounts for a fraction of sporadic ALS cases. Its pathomechanism is incompletely understood. Previous studies have linked mitochondrial defects and ROS to pathogenesis in fly, iPS, mouse, etc. models, and antioxidants can suppress some neurodegenerative features in these models. Consistent with these findings, one of the only two effective drugs for general ALS, edaravone, is believed to mitigate oxidative stress in motor neurons. Hence, oxidative stress is a critical pathogenic contributor that holds great potential as a therapeutic target. However, our understanding of its cause and consequence in ALS is limited. This paper includes at least two novel points: 1) identifying mitochondrial, but not cytosolic, ROS is upregulated and contributes to neurodegeneration in C9ALS models; 2) discovering that the Keap1/Nrf2 is altered and activating Nrf2 suppresses neurodegeneration. The first point presents an incremental advance in the field, but the second one is potentially critical, especially from a translational aspect. That being said, the novelty of the second point is somewhat dampened by a recently published paper (Jiménez-Villegas, et al. 2022), which showed that Nrf2/Keap1 is altered in C9 patient leukocytes and NSC cells overexpressing or treated with C9-DPRs. However, these cells/models are remotely related to the disease. The current manuscript still provided evidence in an in vivo neuronal model for the first time. If the authors could make their iPS part comprehensive, this could still be a major advance towards translation.

This paper could be interesting to a broad audience beyond the ALS field.

Another strength of this paper is that the fly analyses are comprehensive, the data are convincing, and the conclusions are solid. However, the major weakness is that the iPSN part is incomplete to support the translatability of their findings in flies. Current data only suggest that DMF and EDV are functional in iPSNs.

Reviewer #2

Evidence, reproducibility and clarity (Required):

the study of ALS uses almost exclusively drosophila larvae and adults and has a few expts with iNeurons (human) at the end. The results are interesting and relevant to human disease and do suggest potential ways to treat disease. Not all the effect sizes are large, but nonetheless this is publishable material. More expts would of course strengthen their case. None of what I suggest is essential, but this depends in part on where they eventually want to publish their work.

Some comments below:

1. All are overexpression models with strong phenotypes. This has to be mentioned.

- The nature of the genetic models is clearly delineated in the manuscript. To highlight this further in the text, we have added comments at the start of the Results section stating that *Drosophila* do not have an orthologue of *C9orf72*, so we use previously established transgenic models (lines 372-376). In fact, it is incorrect to call these 'overexpression' models because there isn't a *C9orf72* orthologue to be overexpressed. Formally, they are ectopic expression models.

2. Furthermore, in any ageing model every aspect of cell biology is affected.

- Agreed.

3. In fig 1E to the non-expert it is hard to work out what is a mitochondrion. Some higher res imaging might help.

- It is indeed difficult to discern individual mitochondria with this particular approach. We have a lot of experience in this kind of analysis and higher resolution imaging does not resolve the problem. The challenges with imaging mitochondria in such tiny cell bodies is the reason that we have adopted a categorical scoring system.

4. Line 390 comments on morphology but fig s1b-c is survival. Do they have morphology data? If not then they should rephrase the text

- This is a misunderstanding. The brief mention of mitochondrial morphology at the start of the paragraph ("*Mitochondrial morphology is known to respond to changes in reactive oxygen species (ROS) levels as well as other physiological stimuli.*" – lines 414-415) is to provide as a segue from the preceding section describing the morphology defects to the following sections that investigate the possible mechanisms affecting this.

5. Line 441. Can they provide reference for 1000 being physiologically relevant? 36 is certainly pathological in humans. In my opinion the only genuinely physiologically relevant model is a genetically faithful knockin without codon alteration.

- We have rephrased this to be 'more physiologically relevant repeat length' and provided a reference.

6. Line 482 - they say mitophagy is downstream, but isn't that obvious in a C9 transgenic model?

- We appreciate that this statement was confusing. We are referring to 'upstream' or 'downstream' in the cascade of events that ensue from expression of DPRs, not upstream or downstream with respect to C9 mutations themselves, so we have rephrased this as "not a primary contributor to *C9orf72* pathology" (lines 502-503).

7a. Line 502 - they indicate 'exploring the basis', but I am a little unclear what they are saying. What is the reason for the reduced SOD1 in x36 v x3 flies? Are they simply killing cells that have the most SOD1 and therefore their qPCRs/blots only represent those cells with less SOD1? There is still SOD1 being expressed there of course.

- Thanks for allowing us to clarify this point. We have not been able to clarify the mechanism for why *Sod1* appears to be downregulated upon G4C2x36 expression, which we acknowledge is a limitation. So, we have decided to adjust the language from 'exploring the basis', to now simply report this as an associated observation (line 527).

7b. In the text it would help if they clarified if the genes overexpressed are human or fly. If human, it might be worth overexpressing mutant ALS SOD1 if they are able.

- In general, when reporting on experiments with a model organism such as *Drosophila*, we work on the assumption that genetic manipulations will typically be that of the host species, i.e., transgenic expression with

be of *Drosophila* genes, unless specifically stated otherwise. In any case, all the necessary details of all genetic strains used in this study are laid out in Methods.

8. Line 521 - this para should perhaps be in intro section, not results.

- Agreed. We have now edited the start of this section (lines 543-546).

9. In Fig5, do they have CnnC IHC to back up their conclusion that *Keap1* mutation is affecting this process?

- Thank you for this suggestion. We have now analysed CncC localisation in C9 models \pm *Keap1* mutation. As before, we saw that G4C2x36 caused an increase in CncC nuclear localisation, although there was a trend towards an increase with *Keap1* heterozygosity this was not consistent enough to be significant. These data are presented in new Fig. 5D, E and discussed in the text (lines 579-581). Although these results do not show an additional increase of nuclear CncC by this treatment of DMF, we also performed qRT-PCR analysis of CncC target genes *GstD1*, *GstD2*, *Gclc* and *Cyp6a2*, from flies treated with or without DMF. These data show that the degree of transcriptional activation was variable between different targets, but DMF treatment caused a general upregulation of CncC targets in G4C2x36 flies (new Fig. 6A).

10. The Induced neuron results are interesting. What kind of neurons are they? Have they been confirmed to be so with ICC? The figures in 6 are poor. They should make the point that correction of the mutation to ensure isogenicity would be an additional confirmatory measure. Isogenic lines are available from JAX and the UK MND Institute.

- Agreed. We now provide further characterisation of the iNeurons that was done at the time of the original experiments but not presented. These analyses include immunostaining with neuronal marker antibodies against β -III Tubulin, MAP2 and NeuN. These data are shown in new Supplementary Figure 3A, B. We also report the relative viability of these neurons at the point of analysis (new Supplementary Figure 3C, D). We have added mention of this in the text (lines 620-621 and 627-628). Of note, these patient cell lines have been used and reported before (Reference 53) which we cite on line 618. We also acknowledge the limitations of using these lines, and that future work would be better done with isogenic controls (lines 690-692) as the reviewer indicates.

11. Suppl fig 3 - interesting observation with edaravone, but do they have any survival/motility data in neurons/flies? Also, would be good to compare with another drug that works on a different mechanism E.g. riluzole.

- Since edaravone is a known therapeutic for ALS and was used as a comparator, rather than being the primary focus, we do not have additional data on edaravone.

12. Overall, the conclude they have done a comprehensive analysis of mito function, but I would argue that while a good analysis there are plenty of other studies they could have done e.g. assess mitochondrial respiratory chain.

- We agree that additional studies can always be envisaged.

13a. I also think the imaging of mitochondria could be better, and much work needs to be done on the iNeurons to characterise them.

- As mentioned above, we have provided additional characterisation of the iNeurons in this revision.

13b. Sentence line 674 - needs rephrasing.

- Thanks for prompting this. We have now rewritten these sentences (now lines 700-701).

14. In their final paragraph what do you they mean by oxidative stress being upstream? I would argue it is downstream of the C9 expansion, right?

- We apologise that this was confusingly written. As per the comment above (response to point 6), we were referring to events 'upstream' or 'downstream' in the cascade of events that ensuing from expression of DPRs. We have now rephrased this to be a "proximal" pathogenic mechanism (lines 708-710). We hope that our intended meaning is now clearer in the text.

Significance (Required):

A good study, modest degree of advancement in the field.

Reviewer #3

Evidence, reproducibility and clarity (Required):

In the present paper the authors focused on the hyper-production of ROS in a C9orf72 fly model. they the sought to rescue the observed fly phenotype by manipulating mitochondria dysfunctions or pathways downstream these dysfunctions.

Majors:

1. Given the wide varieties of statistical tests used a rationale should be given to why a certain test (one way anova) was used in one experiment (WB, qPCR) and another for another (Chi square) experiment (mitochondria morphology)

- In all cases, the choice of statistical test is dictated by the nature of the data being analysed – a principal that should be well-understood by all experienced researchers – and so may vary between experiments but will be consistent between different data sets of the same type of experiment. For instance, for those data sets consisting of two groups, an unpaired t-test would be appropriate. Most other experiments consist of three or more experimental groups and so will need an appropriate test with additional post-hoc test to correct for multiple comparisons, such as one-way ANOVA with Bonferroni's post-hoc correction. Where data sets are not normally distributed, such as generated by our climbing assay, a non-parametric analysis is required, such as the Kruskal-Wallis test. Here we also use a Dunn's post-hoc correction for multiple comparisons. In some assays of multiple groups, there are also multiple variables, such as the different drug concentrations tested on control and C9 iNeurons, a two-way ANOVA with an appropriate post-hoc correction test is used. Finally, some assays employ a categorical scored system, such as the mitochondrial morphology analysis, which will require a different type of statistical analysis such as Chi squared test.

These types of analysis are in no way unusual or 'cherry-picked' to give the most desirable outcomes but are selected simply based on the type of the data to be analysed following standard rules of statistical analysis. For this reason, we do not feel that any more elaborate explanation is necessary in the manuscript text itself, but we hope that the explanation given here will satisfy the reviewer of the rationale for employing different statistical tests for different data sets.

2. The entire second part of the paper, and most important one to the authors (given the tile), rely mostly on a supposed loss in protection against antioxidant. I feel the experiment in support of this hypothesis are not strong. It is true that there is an overproduction of ROS (as evaluated in the first figures) but the loss in protection stated based on Fig 4H does not hold much. I think more experiment are needed to support this hypothesis.

- This is a fair comment and on reflection we also agree that our claim that the response to oxidative stress is blunted in the C9 models is based almost exclusively on the data from (old) Fig. 4H, and so is not strong. On reflection, prompted by the reviewer's comment, we have removed this interpretation from the manuscript and revised our comments accordingly. Consequently, we have also removed Fig. 4H.

3. Moreover, I counter intuitive that to rescue a phenotype the authors over expressed that is already high in C9orf72 flies (nrf). I would suggest to match this results with downregulation of nrf, to effectively proof that nrf decrease is detrimental to counteract ROS species in C9orf72 flies (further reducing protection against ROS). I believe this experiment is quite critical for the entire manuscript.

- We appreciate the thinking behind this suggestion, but this experiment can't be performed because loss of CncC function is lethal, as expected from a master regulator of a major cell-protection mechanism.

4. Also to me there is a little bit of disconnection between the first three figures and the last three. The authors also find a reuse effect over expressing SOD2 etc as shown in figure 3 where they actually show rescue in mitochondrial dysfunction (morphology etc). The only piece of data that shows rescue in mitochondrial dysfunction upon nrf over expression is figure 5H. More extensive characterization of mitochondrial dysfunction recur should be performed if the title want to kept focused on keep/nrf mechanism. Otherwise a broader title like "modulation of the mitochondria damage rescue C9orf72 phenotype." could help the reader understanding the overarching message of the paper

- We do not see a disconnect between the first part of the paper and the second. To be clear, the first part was documenting mitochondria-related defects (morphology, ROS, mitophagy) and determining their causative hierarchy and mechanistic impact on organismal phenotypes (we found only certain antioxidants rescued locomotor deficits and could reverse mitochondrial morphology and mitophagy defects). As stated, these results strongly implicated oxidative stress as a major driver in organismal pathology. The second part of the study was characterising whether a major antioxidant defence pathway (Keap1/Nrf2) could be manipulated to provide phenotypic rescue on the organismal scale (i.e., locomotor behaviours). On reflection of the original title, we agree that this was too focussed on the mitochondrial dysfunction angle (and also gave too much prominence to the iNeuron part of the study). Therefore, we have now modified the title to reflect a greater focus on oxidative stress and locomotor behaviours across the study. We hope this the reviewer feels that this better represents the study but will be happy to consider suggested alternatives.

Minors:

5. Figure 1n does each for represent a cell? or is an average of more cells and each dot represent an animal? I could not find this information anywhere, but if each dots is a single cells, I would recommend scaling up to at least 10 cells. Same concern for Figure 3F

We agree that this point needs clarification. Each dot represents data for one animal. The quantification per animal is based on at least 10 cells from one image. This has been added to the Methods section for clarification (lines 220-221).

6. Line 550-1-2 I do not agree with the statement. I do not think that the data shown that the protection against ross is less efficient. The only difference is the starting point. But the final point is the same so why should protection against ROS be less efficient in G4C2x36 drosophilas?

- This comment relates to point 2 above. As stated there, we agree that the data are not compelling enough to make this interpretation, so we have revised our comments accordingly.

7. There are some concerns about the neurons in figure 3: they do not appear to have axons and dendrites. I'd suggest containing with neuronal marker.

- The reviewer may be unfamiliar with the specific tissue in question; the larval ventral ganglion. As a complex, mature tissue there are multiple cell types (e.g., neurons and glia) very closely packed. Neuronal processes are very thin in this tissue, and they are squeezed between neighbouring cells. Thus, microscopy of neuronal cell biology within such a complex tissue does not look like in vitro cultured neurons. In the specific context of Figure 3, we are looking at markers for mitochondria or mitophagy. The reviewer may also be aware that mitochondria and mitolysosomes are most abundant in the cell bodies and have very limited abundance in neuronal processes. Thus, we do not generally try to observe these organelles in processes because there

would be very little to see. We know that the signal is within neurons because the markers are transgenically expressed exclusively by a neuronal driver system i.e. *nSyb*-GAL4. In summary, there is no problem with how these cells or how they look. This is quite normal.

8. iNeurons were only used to confirm the second part of the paper. Would be interesting to also confirm some of the results in the first part, like SOD2 over expression etc etc.

- We appreciate this suggestion, which is similar to a comment from Reviewer 1, but, as replied above, time, personnel and resource constraints preclude additional investigations on this occasion. Just to reiterate, it is worth noting that the cell models were used specifically to validate that elevated mitochondrial oxidative stress and increased nuclear Nrf2 localisation also occurred in patient-derived neurons, and whether DMF treatment could reverse the oxidative stress. This was the extent to which the cell models were used in this instance and the current data are sufficient to support the conclusions made based on this. We regret that it was not possible to delve deeper into this at the current time but would be the focus of future work.

Significance (Required):

The present work while not extremely novel in the hypothesis, it is well performed with state-of-the-art techniques, some of them also very novel to the field. The concept of oxidative stress as an important in ALS pathogenesis is not new in the field, but the identification of Nrf as an important players might pave the way for more human related studies and possibly to therapeutic interventions.

I think the work is technically sounded and well performed; certain evidence are solidly demonstrated with multiple different techniques. other evidences instead need a little more work to prove their solidity to widen the audience which will appreciate the content of this paper.

May 31, 2024

RE: Life Science Alliance Manuscript #LSA-2024-02853-T

Dr. Alexander J Whitworth
University of Cambridge
MRC Mitochondrial Biology Unit
Hills Road
Cambridge CB2 0XY
United Kingdom

Dear Dr. Whitworth,

Thank you for submitting your revised manuscript entitled "Activation of the Keap1/Nrf2 pathway suppresses mitochondrial dysfunction, oxidative stress and motor phenotypes in C9orf72 ALS/FTD models". We would be happy to publish your paper in Life Science Alliance pending final revisions necessary to meet our formatting guidelines.

- please be sure that the authorship listing and order is correct
- please upload all figure files as individual ones, including the supplementary figure files
- please add a Running Title and a Summary Blurb/Alternate Abstract to our system
- please add a Category for your manuscript in our system
- please add the Twitter handle of your host institute/organization as well as your own or/and one of the authors in our system
- please consult our manuscript preparation guidelines <https://www.life-science-alliance.org/manuscript-prep> and make sure your manuscript sections are in the correct order
- please add author contributions to the system as well. Please note that they need to match with the manuscript file, and that contribution has to qualify the Author for the authorship according to ICMJE guidelines. Funding acquisition, for example, listed as an activity only (without other contributions) does not qualify a contributor for authorship according to ICMJE guidelines
- please add your main, supplementary figure, and table legends to the main manuscript text after the references section

LSA now encourages authors to provide a 30-60 second video where the study is briefly explained. We will use these videos on social media to promote the published paper and the presenting author (for examples, see <https://docs.google.com/document/d/1-UWCfbE4pGcDdcgzcmiuJl2XMBJnxKYeqRvLLrLS08s/edit?usp=sharing>). Corresponding or first-authors are welcome to submit the video. Please submit only one video per manuscript. The video can be emailed to contact@life-science-alliance.org

A. FINAL FILES:

B. MANUSCRIPT ORGANIZATION AND FORMATTING:

Sincerely,

June 4, 2024

RE: Life Science Alliance Manuscript #LSA-2024-02853-TR

Dr. Alexander J Whitworth
University of Cambridge
MRC Mitochondrial Biology Unit
Hills Road
Cambridge CB2 0XY
United Kingdom

Dear Dr. Whitworth,

Thank you for submitting your Research Article entitled "Activation of the Keap1/Nrf2 pathway suppresses mitochondrial dysfunction, oxidative stress and motor phenotypes in C9orf72 ALS/FTD models". It is a pleasure to let you know that your manuscript is now accepted for publication in Life Science Alliance. Congratulations on this interesting work.

DISTRIBUTION OF MATERIALS:

Again, congratulations on a very nice paper. I hope you found the review process to be constructive and are pleased with how the manuscript was handled editorially. We look forward to future exciting submissions from your lab.

Sincerely,
